# The Long-term Transport and Radiative Impacts of the 2017 British Columbia Pyrocumulonimbus Smoke Aerosols in the Stratosphere

Sampa Das[1,2], Peter R. Colarco[1], Luke D. Oman[1], Ghassan Taha[1,3] and Omar Torres[1]

[1]NASA Goddard Space Flight Center, Greenbelt, Maryland, USA
[2]Universities Space Research Association-NASA Postdoctoral Program, Columbia, Maryland, USA
[3]Universities Space Research Association, USRA, Greenbelt, Maryland, USA

*Correspondence to*: Sampa Das (sampa.das@nasa.gov)

**Abstract.** Interactions of meteorology with wildfires in British Columbia, Canada during August 2017 led to three major pyrocumulonimbus (pyroCb) events that resulted in the injection of large amounts of smoke aerosols and other combustion products at the local upper troposphere and lower stratosphere (UTLS). These plumes of UTLS smoke with elevated values of aerosol extinction and backscatter compared to the background state were readily tracked by multiple satellite-based instruments as they spread across the Northern Hemisphere (NH). The plumes resided in the lower stratosphere for about 8-10 months following the fire injections. To investigate the radiative impacts of these events on the Earth system, we performed a number of simulations with the Goddard Earth Observing System (GEOS) atmospheric general circulation model (AGCM). Observations from multiple remote-sensing instruments were used to calibrate the injection parameters (location, amount, composition and heights) and optical properties of the smoke aerosols in the model. The resulting simulations of three-dimensional smoke transport were evaluated for a year from the day of injections using daily observations from OMPS-LP (Ozone Mapping Profiler Suite Limb Profiler). The model simulated rate of ascent, hemispheric spread and residence time of the smoke aerosols in the stratosphere are in close agreement with OMPS-LP observations. We found that both aerosol self-lofting and the large-scale atmospheric motion play important roles in lifting the smoke plumes from near the tropopause altitudes (~12 km) to about 22-23 km into the atmosphere. Further, our estimations of the radiative impacts of the pyroCb-emitted smoke aerosols showed that the smoke caused an additional warming of the atmosphere by about 0.6-1 W/m$^2$ (zonal mean) that persisted for about 2-3 months after the injections in regions north of 40$^o$N. The surface experienced a comparable magnitude of cooling. The atmospheric warming is mainly located in the stratosphere, coincident with the location of the smoke plumes, leading to an increase in zonal mean shortwave (SW) heating rates of 0.02-0.04 K/day during September 2017.

## 1    Introduction

When convective smoke plumes from large wildfires are intercepted by favorable meteorological conditions, such as those that produce dry thunderstorms (Peterson et al., 2017), the formation of  fire-triggered thunderstorms, called pyrocumulonimbus (pyroCb, Fromm et al., 2010), can occur. In extreme cases, pyroCbs release copious amounts of smoke and other combustion products into the upper-troposphere-and-lower-stratosphere (UTLS). Due to less efficient wet and dry removal processes for aerosols at these altitudes, the resulting aerosol particles can persist

for much longer times and be carried over much longer distances (~months, globally) compared to cases where aerosols are injected into the lower troposphere and boundary layer (~days, hundreds of km).

       A major pyroCb event occurred in British Columbia (BrCo), Canada, in August 2017. While pyroCb events are not rare occurrences for the mid- to high-latitude regions during the dry summer seasons (Peterson et al., 2016), this was the largest known stratospheric intrusion from pyroCb activity at the time, with aerosol injection amounts

estimated at 0.1–0.3 Tg (Peterson et al., 2018) and 0.18 – 0.35 Tg (Torres et al., 2020), comparable to the aerosol produced in a moderately sized volcanic eruption. However, unlike aerosols originating from volcanic eruptions that exert an overall cooling effect on the planet due to their predominantly scattering nature (Robock, 2000; Solomon et al., 2011; Vernier et al., 2011), the smoke aerosols from pyroCb events contain black and brown carbon (BC and BrC) particles that strongly absorb incoming solar radiation and thus warm the surrounding atmosphere. This atmospheric

heating by smoke can lead to an overall positive or negative effect on the radiation balance at the top-of-the-atmosphere (TOA) depending on the aerosol plume's vertical location (Ban-Weiss et al., 2012), its mixing state (Jacobson, 2001) and the albedo of the underlying surfaces (Boucher et al., 2013; Keil and Haywood, 2003).

       The smoke from the August 2017 BrCo pyroCbs resulted in enhanced aerosol extinction and backscatter in the UTLS that were significantly above the values in clean background conditions. The plumes from these pyroCbs were

readily tracked by satellite-based remote sensing instruments (Khaykin et al., 2018; Lestrelin et al., 2020; Torres et al., 2020) and ground-based lidar networks (e.g., Ansmann et al., 2018; Baars et al., 2019) as they spread across the Northern Hemisphere (NH), and were observed to persist in the stratosphere for about 10 months following initial injections. Observations from the space-based CALIOP (Cloud-Aerosol Lidar with Orthogonal Polarization) lidar showed that optically thick smoke plumes rose from their ~12 km injection altitude to an altitude of ~22 km within

19 days, with an especially steep ascent rate of 2-3 km per day in the first few days after the injection (Khaykin et al., 2018). Torres et al. (2020) used data from both the Earth Polychromatic Imaging Camera (EPIC) sensor and the Ozone Mapping and Profiler Suite (OMPS) Limb Profiler (LP) to observe the time evolution of the pyroCb-emitted plumes. High ultraviolet aerosol index retrievals (~24-29) from EPIC's near-hourly observations in the week following the injections were used to retrieve lofting of the smoke plume consistent with the CALIOP observations (Torres et al.

2020), while OMPS-LP showed that the aerosol extinction persisted above the clean background stratosphere levels for a 10-month period between August 2017 and June 2018. Torres et al. (2020) postulated aerosol self-lofting—that is, increased buoyancy in the smoke plume brought on by heating in the plume by absorption of solar radiation—as the determinant mechanism for the initial rapid ascent of the plume. This hypothesis was supported by model experiments that are also the subject of this paper. Kloss et al. (2019) used satellite observations and models to further

highlight the role of the Asian summer monsoon anticyclone (ASMA) in additional lofting of the smoke beyond 18 km as it was transported over the tropical UTLS.

       The aerosol self-lofting mechanism postulated in Torres et al. (2020) has previously been discussed for optically thick smoke plumes (Herring and Hobbs, 1994; Malone et al., 1986; Radke et al., 1990). In the context of wildfire-induced pyroCbs, this mechanism was first postulated by de Laat et al. (2012), who modeled the lofting of the smoke

produced in the 2009 Australian 'Black Saturday' fires. Using one-dimensional plume height radiative transfer

calculations based on Boers et al. (2010) and observation-based assumptions for the aerosol optical properties and dynamical conditions, they showed a plume rise of 10 km within 3 days of the initiation of this event. Their study, however, was limited in that they lacked a full accounting of the impacts of aerosol-radiation and dynamical coupling that a 3D chemistry-climate model can provide. Our study here addresses this limitation.

With respect to the specific BrCo pyroCb event we highlight two recent modeling studies. Christian et al. (2019) simulated this event in an offline global chemical transport model (CTM) and provided the resulting aerosol distributions as input to a radiative transfer (RT) model in order to obtain estimates of aerosol direct radiative forcing due to the pyroCb smoke. While they accurately modeled the integrated aerosol lifetime and initial transport of the smoke in the atmosphere, their model was unable to simulate the observed longer-term aerosol transport over the
tropical UTLS that occurred several weeks after the event. The observed plume rise rate, hemispherical spread, and stratospheric lifetime were also accurately simulated by Yu et al. (2019), who used a radiatively and chemically interactive atmospheric general circulation model (AGCM) coupled to a detailed aerosol microphysics code. Their results were in close agreement with SAGE-III (Stratospheric Aerosol and Gas Experiment III) observations and suggest a complex morphology for smoke particles, where BC represented as fractal aggregates with a non-spherical
coating of organic carbon (OC) was necessary to impart the needed radiative heating to loft the smoke as observed. In addition, they included a mechanism for photochemical loss of organics within the smoke via stratospheric ozone to better match the SAGE-III observed decay of the pyroCb smoke plumes in the stratosphere.

We show here for the first-time model simulations of the three-dimensional transport of the smoke following the August 2017 BrCo pyroCb event that show excellent agreement with the OMPS-LP observations in terms of both
the near-field, self-lofting driven vertical ascent of the smoke following its injection, as well as its longer-range dynamical interaction with the ASMA. Our model is intermediate in complexity between the models used in the Christian et al. (2019) and Yu et al. (2019) studies. Here we used the Goddard Earth Observing System (GEOS) Earth system model, which includes aerosol and chemistry mechanisms coupled to the underlying AGCM physical and dynamical cores. The dynamics in our simulations are constrained by assimilated meteorology provided by the
Modern-Era Retrospective analysis for Research and Applications, Version 2 (MERRA-2; Gelaro et al. 2017), which is by design a strong constraint in the troposphere but here is relaxed in the UTLS and higher altitudes to allow our radiatively coupled aerosols to influence the atmospheric circulation resulting from the pyroCb event. Retrievals of smoke aerosol properties from multiple remote-sensing instruments were used to calibrate the injection location, timing, amount, and optical properties of the smoke aerosols. The resulting "best-estimate" simulation of smoke
transport was evaluated over a year using observations from OMS-LP, which has a higher temporal resolution compared to SAGE-III and has a better sensitivity to the stratospheric aerosols than CALIOP.

The paper is organized as follows. Section 2 briefly describes the GEOS model and the specific model configuration we used for this study, along with a brief description of the observational datasets we used for model calibration and evaluation. Section 3 discusses the results of the comparative analysis between model simulated three-
dimensional plume transport and the OMPS-LP observations. The section further discusses the impacts of pyroCb smoke aerosols on atmospheric and surface radiative forcing and on the perturbations in stratospheric heating rates.

We also put our findings into perspective by comparing our key model assumptions with previous modeling studies. Finally, Section 4 summarizes the major conclusions of the study.

## 2    Approach and Methods

### 2.1    Model Description and Configuration

The NASA Goddard Earth Observing System (GEOS) system is a weather-and-climate capable Earth system model consisting of components for atmospheric circulation and composition, oceanic circulation and biogeochemistry, land surface processes, and data assimilation (Molod et al., 2015; Rienecker et al., 2008). We used the GEOS AGCM to simulate the transport and subsequent impact of the three major pyroCb-triggered smoke aerosol

injections over BrCo in August 2017 (Fig. 1a; Peterson et al. 2018). The GEOS AGCM can be run primarily in two modes: free-running and replay. The free-running mode is the typical climate model configuration, where the model integrates forward in time from a given set of initial conditions, either with prescribed sea surface temperatures as a lower boundary condition or else with a coupled ocean model. The replay mode, on the other hand, mimics the atmospheric data assimilation step taken in most atmospheric forecasting systems, by using prescribed meteorological

fields (i.e., temperature, pressure, horizontal winds, and specific humidity) from a prior atmospheric analysis to constrain the simulated meteorology via an incremental analysis update. In the replay, the full model physics is still run every time step, but the model response is only weakly impacted by internal forcings that arise from, for example, the radiative impacts of strong aerosol events.  In this way the replay provides a capability like that of a traditional chemical transport model (CTM) and a way to simulate real events at only a fraction of the computational cost of

rerunning the full data assimilation system. We performed a number of simulations in replay mode with varying injection altitudes. However, for all such simulations, soon after about a week from the injections, the horizontal transport pattern of the smoke plumes started to deviate from the observations and majority of the simulated smoke plumes ended up close to the Arctic, instead of being transported towards the tropics based on the observations from multiple satellite instruments. Similar smoke transport pattern was reported in Christian et al. (2019) that used a CTM

set-up to perform their pyroCb simulations. This anomalous model behavior prompted the need to precisely simulate the rate of ascent of the smoke plumes resulting from aerosol self-lofting, since horizontal transport is closely tied to the vertical location of the smoke plumes. Therefore, for the simulations of this study, we modified the replay settings in the model to allow for temperature (T) and specific humidity ($Q_v$) blending at levels around the modeled tropopause such that the simulated T and $Q_v$ are not adjusted towards their reanalysis values in the stratosphere, but continue to

be adjusted in the troposphere. This modification allowed the stratospheric temperature changes due to aerosol heating to remain unaltered when the model adjusted to reanalysis fields every 3 hours, thus aiding in vertical transport of the pyroCb plume in the stratosphere through aerosol self-lofting. Simultaneously, large-scale horizontal plume transport was still guided by the reanalysis winds similar to a regular replay model run.

The prognostic aerosol module within the GEOS AGCM is based on the Goddard Chemistry, Aerosol,

Radiation, and Transport module (GOCART, Chin et al., 2002, 2009; Colarco et al., 2010). GOCART simulates seven tropospheric aerosol species: black carbon (BC), brown carbon (BrC), organic carbon (OC), nitrates ($NO_3$), sulfates

(SO$_4$), dust, and sea salt. For biomass burning emissions, all of the organic carbon mass is accounted within the BrC component, whose optical properties were adjusted to represent a 100% internal mixture of OC and BrC (Hammer et al. 2016; Colarco et al., 2017) that has an enhanced absorption at near-UV wavelengths compared to weak and spectrally flat absorption of traditional OC. The optical properties of other aerosol species are primarily prescribed using the OPAC data set (Hess et al., 1998), except for dust. Dust optics were updated in the model following Colarco et al. (2014). The seven aerosol species are treated as external mixtures that are transported online and radiatively coupled with the GEOS AGCM. The loss processes include wet scavenging and dry deposition. The wet scavenging consists of both scavenging in convective updrafts and rainout/washout in large-scale precipitation. Dry deposition includes gravitational settling as a function of aerosol particle size and air viscosity and surface deposition as a function of surface type and meteorological conditions (Chin et al., 2004).

The model experiments were designed using the Icarus 3.3 version of the GEOS system and were run on a cubed-sphere horizontal grid at ~50 km horizontal resolution with 72 hybrid vertical sigma levels extending between the surface and 0.01 hPa (about 85 km). The hybrid coordinate system is terrain following near the surface and becomes pressure following at higher altitudes (near 180 hPa). The model includes a comprehensive set of physical parameterizations for moist processes, longwave and shortwave radiation, turbulence, land-surface processes, and gravity wave drag (Molod et al., 2015; Nielsen et al., 2017). The moist physics module contains parameterizations for convection using the Relaxed Arakawa-Schubert scheme (Moorthi and Suarez, 1992) and a single-moment parameterization for large-scale precipitation and cloud cover described in Bacmeister et al. (2006). Relevant to our study, note that aerosols are radiatively interactive with the clouds in the model, and therefore impacts of underlying clouds on atmospheric heating by absorbing aerosols are inherently accounted for in the model. The meteorological fields for the model restarts and replay were obtained from version 2 of the Modern-Era Retrospective analysis for Research and Applications (MERRA-2, Gelaro et al. 2017). The stratospheric chemistry component of GEOS, StratChem (Considine et al., 2000; Douglass and Kawa, 1999) was used and provides a simulation of the background stratospheric sulfate aerosol, similar to what is used in Chen et al. (2018).

While we discuss below and performed a number of simulations to calibrate our model, the final results presented are mainly from two model experiments that have a similar setup and were designed specifically to quantify the impact of pyroCb generated stratospheric aerosols on the atmosphere. The main experiment (referred to as the pyroCb experiment) includes our "best-estimate" injection parameters for the pyroCb event (see below). A separate control experiment (called CTL) is configured identically except it does not include the injection of the pyroCb.

For the pyroCb experiment, we performed several simulations using the GEOS set-up alluded to above to obtain a "best-estimate" for pyroCb injection parameters such that the rise and transport of the model simulated plume following the injections were in close agreement with observations from different satellite-based instruments, primarily OMPS-LP. We discuss the results of model sensitivity to different assumptions of injection parameters that provided the basis for our final choice of injection parameters further in Section 3.2. Based on our "best-estimate" simulation, the values for our injection parameters are as follows. The total aerosol (BrC + BC) emissions from the pyroCb events in our model were at the upper limit (300 kt) of reported satellite-based injection estimates (Peterson

et al., 2018), of which BC mass contributed to about 2.5% (7 kt) of the mass. Since the model cannot explicitly simulate the wildfire dynamics associated with a fine scale event such as this, the emissions were horizontally smeared in 2° x 2.5° latitude-longitude grids around three locations in British Columbia (Fig. 1a). The injections were initialized on August 13, 2017, for a total of 5 hours, 0-3 UTC for the first two sources, and 4-6 UTC for the third source. The injection timings were inferred using the observations of cloud-top brightness temperatures from the Geostationary Operational Environmental Satellite (GOES). Vertically, we injected the smoke aerosols uniformly between 10-12 km, which is comparable to the 11-12.5 km estimate of injection heights derived from the satellite-retrieved cloud-top temperatures and radar measured thermodynamic variables in Peterson et al. (2018). The vertical resolution of the model is ~1 km near the tropopause, similar to other models simulating this event (e.g., Yu et al. 2019, Christian et al. 2019), and the emissions effectively get vertically smeared between about 9-13 km (Fig. 1b) such that while some of the injected mass is certainly in the lower stratosphere the bulk is injected into the upper troposphere.

The pyroCb-sourced aerosols were emitted in addition to the nominal GEOS emission inventories, including biomass burning sources provided by the Quick Fire Emissions Dataset 2 (QFED2) biomass burning inventory (Darmenov and da Silva, 2015). The double-counting of smoke aerosol emissions due to addition of biomass burning sources from QFED2 over the PyroCb locations can be neglected because QFED2 emissions from August 13 around pyroCb locations are small (only ~ 50 kt) compared to the 300 kt of pyroCb-sourced aerosols. QFED2 emissions are injected only within the model-simulated boundary layer and so will not spatially evolve coherently with the pyroCb event in any case.

## 2.2    OMPS-LP Extinction

The Ozone Mapping and Profiler Suite (OMPS) Limb Profiler (LP) instrument onboard the Suomi National Polar-orbiting Partnership (Suomi NPP) spacecraft images the Earth's limb by pointing aft along the spacecraft flight path. The sensor employs 3 vertical slits separated horizontally to provide near-global coverage in 3-4 days. In this study, we use the OMPS version 1.5 aerosol extinction profiles at 675 nm, which are available as a gridded product for 1.5° latitude by 20° longitude horizontal resolution at a daily interval. The vertical resolution of the extinction product is 1 km, extending from 10 to 40 km ASL. The operational cloud screening algorithm in use often flags fresh volcanic and pyroCb plumes as clouds (Chen et al., 2016). Therefore, we used cloud-unfiltered data to ensure that potential biomass and volcanic aerosol signals are apparent in the dataset. Measurements at or below the tropopause are however often affected by cloud contamination. The tropopause altitude is provided by the NASA Global Modeling and Assimilation Office (GMAO). The OMPS-LP version 1.5 algorithm used in this study has been calibrated using realistic stratospheric particle sizes in its retrieval (Chen et al. 2018) and has been extensively evaluated with SAGE-III observations (below) and shows good agreement with the SAGE-III dataset for this event (Chen et al., 2020).

## 2.3    SAGE-III Extinction and CALIOP Attenuated Backscatter

The SAGE-III instrument is mounted on the International Space Station (ISS). Version 5.1 data from the instrument is available from June 2017 onwards and was used in our analysis below. SAGE-III uses solar and lunar

occultation and limb-scatter to infer profiles of trace gases like ozone and aerosol extinction coefficient at nine wavelengths between 384 and 1544 nm. SAGE-III provides a nearly direct extinction measurement in its occultation mode, but the occultation measurement provides generally poor spatial coverage, making measurements only during the sunrise and sunset of each orbit. Thus, SAGE III acquires 30 sets of profiles per day in two latitudes bands which roughly span 60°N to 60°S over the course of a month, with best spatial coverage in the mid-latitudes (30-60°).

CALIOP is a lidar system onboard the CALIPSO (Cloud-Aerosol Lidar and Infrared Pathfinder Satellite Observations) satellite that crosses the equator in the early afternoon around 1330 local solar time (LST) in ascending orbit and at 0130 LST in the descending node, with a 16-day repeat cycle. CALIOP measures both the parallel and perpendicular component of the backscattering signal at 532 nm and the total backscatter at 1064 nm. The measurements are made at a very fine vertical resolution of 30 m within the troposphere that expands to 60 m above 8.3 km (Hunt et al., 2009; Winker et al., 2009). For this study, we have used the Version 4.10 of CALIOP Level 1 total attenuated backscatter profiles at 532 nm.

## 3    Results and Discussions

### 3.1    Calibration of Aerosol Optical Properties

In addition to calibrating of pyroCb injection parameters, we also adjusted the microphysical properties (size distribution and modal radius) of aerosol particles in the model based on other remote sensing observations. Since aerosol optical properties are a function of aerosol microphysical properties, adjustments to the particle size distribution resulted in a new set of assumptions for aerosol optical properties. We made these changes only for the BrC component since it contributes to the majority of the smoke aerosol mass and extinction. Based on the new set of optical properties for BrC (referred as 'pyroCb BrC optics' hereafter), we evaluated our simulated single-scattering albedo (SSA) for the pyroCb-sourced smoke mixture using the observations from multiwavelength ground-based lidars. We systematically discuss the results of our calibration efforts as follows:

**Size Distribution:** The Angstrom Exponent (AE) relates inversely to the average size of the particles and can be derived using aerosol extinction values for a wavelength-pair. SAGE-III retrieves aerosol extinction profiles at multiple wavelengths (385 - 1545 nm) with high precision and accuracy, especially for altitudes between 15-35 km (Thomason et al., 2010; Thomason et al., 2020). Hence, we used SAGE-III retrievals of aerosol extinction at 520 and 1022 nm to derive the AE profiles for the corresponding wavelength pair. The model calibration was performed by adjusting the size distribution and the modal radius of the BrC particles, such that the simulated AE perturbation due to the pyroCb smoke in the stratosphere matches the observations from SAGE-III. Note that during AE analysis below, we exclude the SAGE-III measurements where AE is zero to avoid contributions from clouds in the observations.

Several smoke-influenced stratospheric layers were identified during SAGE-III/ISS overpasses following the pyroCb events, where elevated values of aerosol extinction compared to the background were observed at altitudes above the tropopause. Once identified, the model simulated AE vertical profiles were compared with SAGE-III derived AE profiles at the satellite overpass locations. An example of one such case, representative of the various

instances we evaluated, is depicted in Figure 2. The presence of pyroCb-emitted smoke plume in the lower stratosphere is indicated by the high values of aerosol extinction centered around 14 km, both in the SAGE-III and modeled extinction profiles on September 3 2017 at 42$^o$N and 163$^o$W (Fig. 2a). The corresponding AE profile derived from

SAGE-III (Fig. 2b) shows that for altitudes greater than 20 km that are mostly dominated by background stratospheric aerosols, the AE values are about 2.0. On the other hand, for smoke-influenced layers around 14 km, the relatively lower AE values (~1.3) indicate the presence of larger size particles compared to the background stratospheric aerosols. The modeled AE profile with default assumptions of BrC optics (or size distribution) based on global tropospheric smoke observations (not shown here) was not able to match this contrast in AE values between the

smoke-influenced and background stratospheric aerosol dominated levels.

The larger effective particle size for pyroCb-sourced smoke is possibly due to the rapid coagulation of the individual aerosol particles in dense smoke plumes emitted from extreme pyroCb events. The shifting of the particle size distribution to larger mode diameters and enhancement of particle mass in the accumulation mode for the pyroCb-sourced stratospheric smoke compared to the tropospheric smoke is consistent with the size distribution retrievals of

ground-based lidars (Baars et al., 2019; Haarig et al., 2018). This rapid coagulation of particles soon after their injection was also seen in the modeling results of Yu et al. (2019). Thus, we adjusted the BrC size distribution based on the observational findings. We found that by increasing the modal radius of BrC particles to 0.035 μm from 0.02 μm based on the default BrC optics, we were able to obtain a good agreement between model simulated and SAGE-III retrieved AE profiles for the lower stratospheric levels. The simulated AE profile post-calibration is also depicted

in Figure 2b, wherein the model simulated an AE of ~2.0 at altitudes greater than 20 km and lower AE values of 1.5-1.6 for the smoke-influenced airmasses around 14 km.

**Single Scattering Albedo (SSA):** Plume rise due to aerosol self-lofting is a strong function of the absorption efficiency of the aerosol particles, which is characterized by their SSA assumptions in the model. SSA for a particular aerosol type or component depends on the assumptions of its microphysical properties (e.g., refractive index, size

distribution). SSA is an intensive property for an individual aerosol component, but for an aerosol mixture like smoke, SSA depends on the relative amounts of the aerosol components comprising the smoke and their mixing state. We evaluated the model simulated SSA for the pyroCb-sourced stratospheric smoke plumes using measurements from ground-based Raman lidars (Haarig et al., 2018; Hu et al., 2019) that directly measure aerosol extinction and backscatter.  The Raman lidar observations were taken in Europe (Germany and France), where the smoke from the

BrCo pyroCb event was transported in about 10-15 days after the August 13 injections. First, to confirm the presence of stratospheric smoke layers in the model over lidar observation locations, we show the GEOS simulated aerosol extinction profiles (Fig. 3a) for the model grid closest to Leipzig, Germany on August 22 at 21z, consistent with the observational time and location of Haarig et al. (2018). We also plot here the peak extinction values from the lidar observations for the 15-16 km layer reported in the same study. The model simulated extinction profiles show an

elevated extinction feature at ~15 km, which agrees well with the vertical location of the observed stratospheric smoke plume, but the magnitudes of peak aerosols extinctions are underestimated by the model by about a factor of 5-10. To complement the vertical distribution and to understand the reason for this model extinction bias, we also show the horizontal distribution of aerosol extinctions around the Leipzig, Germany region in Figure 3b, for the same time of

the day and at 15.5 km altitude. The horizontal view shows that Leipzig intercepts only a part of the model smoke plume that has lower extinction magnitudes compared to the bulk of the plume, which is close to Ireland (~55ºN, 5ºW) at the time and has extinction magnitudes closer to the lidar observations. Given the relatively coarser horizontal and vertical resolution of global models in general, this slight displacement or delay of model plumes from point observations, especially during the early period after injections is expected.

Circling back to the SSA comparisons, Figure 3c shows the comparisons of stratospheric smoke SSA between the lidar observations and the model. For the model, we show two sets of SSA results, one for the default BrC optics and other for the pyroCb BrC optics that include the adjusted BrC size distribution as discussed above. It is evident from the comparative analysis (Fig. 3c) that the model simulated SSA for the pyroCb BrC optics case lies within the uncertainty range of the observational SSA at all three wavelengths (355, 532 and 1064 nm) across the spectrum. For the mid-visible wavelength of 532 nm, even though the simulated SSA was close to the upper limit of observational SSA (0.9), the model was able to capture the SSA variation at all the three wavelengths. Overall, the simulated SSA is in better agreement with the observations for the case of pyroCb BrC optics compared to the default BrC optics.

To further utilize the longer time-record of observations from ground-based lidar networks over Europe (Baars et al., 2019), we show the evolution of model simulated stratospheric aerosol optical thickness (sAOT) over Europe (30-60ºN and 20W-40ºE) from the time of initial injections up to the end of 2017 (Fig. 3d). GEOS simulated maximum and mean sAOT over the region are well within the range of AOT magnitudes reported in Baars et al. (2019). For maximum model sAOT in fact, even the rate of decrease of sAOT for the early period is well matched with the lidar data as GEOS simulated values decreased from >0.2 in August to values up to about 0.03 in the beginning of September 2017. For mid-September to December 2017, the mean sAOT remained close to 0.01 in our model, which is slightly higher than the final values (0.002-0.008) reported in Baars et al. (2019).

## 3.2    Optimizing the Smoke Plume Rise

While the total aerosol amount within a smoke plume determines the aerosol extinction or optical depth of the plume, the amount of BC mass within the plume is the primary determinant of the rate of plume rise because of its strong absorbing nature compared to the other aerosol components comprising the smoke. The rise of the pyroCb smoke plumes from the injection levels (~12 km) to higher levels (~22-23 km) in the lower stratosphere in about 20 days was observed by OMPS-LP at a high temporal resolution (Fig. 4, black line). Utilizing this OMPS-LP capability, we tuned our injection parameters, including the BC to BrC mass ratio, such that the model simulations are able to closely match the rate of plume rise based on OMPS-LP observations. It can be noted here that optimizing the smoke plume rise in the model inadvertently optimizes the horizontal transport of the plume, at least for a replay simulation, where the large-scale flow is closely tied to the observationally constrained reanalysis wind fields. This is because the direction of large-scale flow varies with altitude and if the plumes loft too quickly or too slowly, they may be misplaced horizontally, impacting their subsequent spread and residence time at a given altitude. For example, if the smoke plumes loft too quickly the majority of the plume material is transported efficiently poleward, where it is likely to get caught in the descending branch of the Brewer-Dobson circulation (BDC) at this time of the year, and the plumes

move out of the stratosphere too quickly. Therefore, it is critical to optimize the rise of the smoke plumes in the model
prior to estimating the spread and lifetime of the pyroCb smoke in the stratosphere.

To evaluate the rate of rise of smoke plumes in the model (Fig. 4), we compared the model simulated plume tops (colored lines) with the OMPS-LP retrieved plume tops (black line) for about a month following the pyroCb injections in a number of possible configurations. Plume top is defined for both the model and OMPS-LP observations as the first level from the top of the atmosphere (TOA) at which the mean aerosol extinction (averaged over 30-90°N)
is greater than the background extinction. To make a reasonable assumption for background or threshold extinction profile, a 10-day mean extinction profile was computed by averaging over the same latitudinal extent and spanning the period prior to the pyroCb injections (August 1-10, 2017). The different colored lines in Figure 4 show the model sensitivity to the variations in emission injection parameters, to the aerosol-radiative coupling, and to the assumptions of BrC optics in simulating the plume rise. We discuss the results of different sensitivity experiments sequentially.

First, the importance of aerosol-radiation coupling in simulating a reasonable plume-rise rate is demonstrated. When aerosols were considered as passive tracers (magenta line in Fig. 4), plumes lacked the buoyancy induced by aerosol radiative heating. This limited the lifting of the bulk of the aerosol mass across the tropopause boundary after their injections at the upper tropospheric levels (Fig. 1b). Moreover, the further lifting of the small amount of aerosol mass that either crossed the tropopause levels or got directly injected into the stratosphere, was hindered beyond 14-
15 km.  Next, the impact of horizontal distribution of emissions/injections on plume-rise is illustrated. Provided the same injection heights and aerosol mass, the point-source emissions of pyroCb smoke (green line) overestimated the rate of plume rise compared to OMPS-LP observations, while horizontally smearing the emissions over a larger area (2-degree box) provides a good match to the rate of plume rise observed by OMPS-LP. The implication of this overestimate in plume-rise rate for the point-source emissions cannot be judged solely based on Fig. 4 because as
discussed earlier, horizontal transport of the plume closely depends on the rate of plume rise and we find that faster ascent in the case of point-source injections transports the plumes poleward instead of towards the tropics as observed. Finally, we show the impact of our calibration of BrC optics (Section 3.1) on the rate of plume-rise (blue and red line). It appears that the changes in BrC size distribution and the optics have a negligible effect on the rate of plume rise. However, for our "best-estimate" simulations that are evaluated further on, we chose to keep the assumptions of
pyroCb BrC optics such that the model is well-constrained and consistent with observations as closely as possible. Overall, model simulated plumes for the "best-estimate" case (red line) are able to closely match the rate of plume top rise observed by OMPS-LP, apart from the final segment of the plume rise between 20-22 km. The probable reasons for this mismatch in final plume top heights are discussed in the following section.

### 3.3    Plume Transport and the Role of Asian Summer Monsoon Anticyclone (ASMA)

We demonstrated the agreement of our GEOS simulations with CALIOP observations in terms of both horizontal and vertical placement of the pyroCb-emitted smoke plumes in Torres et al. (2020), for a few days following the injections. In Figure 5, we revisit the comparisons of simulated and observed aerosol vertical distributions along CALIPSO satellite tracks on August 13 (night time) and August 14 (daytime) that passed over the injected smoke plumes. In this study, we present a more detailed comparison of simulated aerosol transport for a longer period, but

with OMPS-LP observations that have higher temporal resolution and greater sensitivity to measuring aerosols at UTLS and higher altitudes compared to CALIOP. To this end, we evaluated the simulated transport of the stratospheric smoke plumes at different vertical levels on a daily basis using OMPS-LP observations. Simultaneous matching of horizontal transport pattern and the plume rise rate from previous section provided a robust constraint in our process of model calibration. In about three weeks from initial injections, the smoke plumes rose to their highest levels in the stratosphere, and within a month the smoke plumes spread over most of the NH. Figures 6a-b and supplementary Figures S1a-c demonstrate this plume evolution over the first month at weekly interval.

On the day of the initial injection on August 13 (Fig. 6a), OMPS-LP observations do not show any evidence of pyroCb-emitted smoke plumes at the depicted levels of 16-22 km and neither does the model, but this figure demonstrates the inherent differences in the 'background' state between OMPS-LP and the model prior to the pyroCb perturbation. For the higher levels in lower stratosphere (20-22 km), model slightly underestimates the background aerosol extinctions, especially north of 45°N. In the tropics (0-30°N) tropopause heights are higher (~16 km, Park et al., 2009) compared to the mid and high latitudes, and during the Asian summer monsoon season (June-September), tropospheric trace gases and aerosols are convectively lifted into the UTLS, where they remain largely confined within the transport barriers of the ASMA (Park et al., 2009; Santee et al., 2017; Vernier et al., 2011). During August 2017, the center of the ASMA was in between 15°-45° N and 40° -110° E as defined in Kloss et al. (2019). This explains the enhanced extinction at 16-18 km over south Asia and the eastern Mediterranean for both OMPS-LP observations and the model simulations. However, note that since we used unfiltered data for OMPS-LP for this study, the enhanced extinctions at levels below tropopause (~16 km) in observational panels (Fig. 6a) will most likely include contributions from tropical tropopause layer (TTL) clouds as well.

In Figure 6b, we demonstrate the situation three weeks after the initial injections. The intermediate snapshots in time are provided in the supplementary material (Fig. S1a-b). By September 3, the larger smoke plume from the initial injections broke up into three different vortices (Lesterlin et al. 2020) that have spread over most parts of the hemisphere, especially north of 40°N and at altitudes 16 km and below. For altitudes 18 km and above, smoke plumes traversed along the edges of the monsoon anticyclone, while continuing to lift up to ~22 km in OMPS-LP observation. Clearly, the spatial locations and extinction magnitudes of the GEOS simulated smoke aerosols thus far closely match the OMPS-LP observations. The final model plume ascent, however, falls short by about 1-2 km compared to the OMPS-LP observations, similar to Figure 4. This could possibly be due to a combination of factors. One such possibility is that the model vertical resolution at these altitudes is close to about 1 km, which leads to smearing of the aerosol heating over a larger area and making it critical for further lofting of aerosols in diluted smoke plumes weeks after the injection. Although, it is worth noting that though SAGE-III observations of plume tops over this region and time (Kloss et al. 2019; Yu et al. 2019) are around 20 km ASL, which is more consistent with our model simulations compared to OMPS-LP observations. Resolving the differences between SAGE-III and OMPS-LP in this regard is beyond the scope of this study. Nonetheless, a very small fraction of smoke mass got lofted to levels higher than 20 km (OMPS-LP panels in Fig. S1c; Baars et al. 2019) and thus has a very small contribution to the stratospheric smoke AOD, which is demonstrated in Section 3.5 as well.

### 3.4 Hemispherical spread and Residence Time of the PyroCb Smoke

The spread of the smoke plumes is also relevant in determining the residence time of the stratospheric smoke at observed altitudes. The timeseries of zonal mean stratospheric AOD retrieved from OMPS-LP (Fig. 7a) for about a year shows the significant enhancements in stratospheric AOD values compared to the background state that occur over the mid and high latitudes about 10-20 days after the injections. These enhanced extinctions persist for about 8-10 months after the injection. The comparison with the zonal mean stratospheric AOD obtained from the model (Fig. 7b) shows that the model is able to closely match the latitudinal spread and residence time of the AOD perturbations with OMPS-LP observations. However, subtle differences exist. For example, the largest perturbations in AOD for the model occur immediately after the injection, while for OMPS-LP this is observed only after 10-20 days of injection. It is quite common for limb instruments to underestimate the initial plume extinction magnitudes, as reported for volcanic eruptions as well (e.g., Haywood et al. (2010) and Kloss et al. (2021)). For OMPS-LP in particular, this underestimation is most likely caused by a combination of its coverage, when the plume is not well mixed, and large sampling along the line-of-sight, which is 125 km along-track, and up to 200 km cross track.

Zonal mean total AOD comparisons (Fig. 7) further reveal that there is a slight overestimation of stratospheric AOD by the model compared to OMPS-LP overall. This is possibly due to the inherent differences in background stratospheric aerosol extinctions simulated by the model compared to OMPS-LP. To further investigate this possibility, we derived a reasonable estimate of smoke AOD from OMPS-LP by removing a background value from the daily retrievals of aerosol extinctions for the current year (2017-18). The background value is assumed as the monthly mean aerosol extinction of the previous year (2016-17), since there were no strong volcanic eruptions or pyroCb events over this period in the OMPS-LP observations. For the model however, the stratospheric smoke AOD is simply computed as the differences in stratospheric AOD between the pyroCb and CTL experiments. The comparisons of zonal mean smoke AOD over time between the model and OMPS-LP are depicted in Figure 8. It is clear from the comparisons that compared to the stratospheric AOD, the magnitudes of stratospheric smoke AOD in the model show an overall better agreement with the OMPS-LP observations than for the total stratospheric AOD. This provides the necessary evidence that the differences in stratospheric AOD are actually a result of the differences in background state between the model and the observations.

### 3.5 Vertical distribution of the stratospheric smoke

The PyroCb emitted smoke perturbed the background state of the lower stratosphere, but by different amounts at different vertical levels. Figure 9 shows the vertical distribution of aerosols, where AOD for atmospheric columns extending from the TOA down to different altitudes are compared between OMPS-LP and the model over time. The plots depict an increase in AOD as the plumes ascended to higher levels in the stratosphere, followed by a decrease back to their background state as the smoke aerosols descended with the large-scale circulation and eventually moved out of the stratosphere. Overall, the model is able to closely match the residence time of smoke aerosols at observed altitudes, especially for middle atmospheric levels between 16 and 18 km. For lower levels of 14 km, model AODs are much higher than the observed AODs for the first 10-20 days after the injection, suggesting optically thicker smoke plumes in the model for lower levels compared to OMPS-LP. As stated earlier, this is due to the shortcoming of limb

instruments in general in accurately measuring extinction magnitudes in the early period after volcanic eruptions as well. For the higher levels (>20 km), the negative bias of model simulated AOD is due to a combination of two reasons. First, the model background stratospheric aerosol extinctions are lower than OMPS-LP retrievals at these levels as discussed in Section 3.3. Second, the model smoke plumes never reached as high as 22 km as observed by OMPS-LP.

### 3.6    Smoke Aerosol Impacts on Radiation Balance

Having established the good agreement between the model and OMPS-LP observations in terms of AOD, hemispherical spread and residence time, we used our "best-estimate" simulation to assess the impact of pyroCb-emitted aerosols on Earth's radiation balance. The surface and atmospheric radiative forcing, which is calculated as the difference in TOA and surface radiative forcing were computed for both the pyroCb and CTL experiments. The forcing calculations are based on all-sky and combined longwave (LW) and shortwave (SW) values. The differences in radiative forcing calculations between the two experiments represent the pyroCb-caused perturbation and are presented as zonal means in Figure 10.  We found that the presence of pyroCb aerosols caused a warming of the atmosphere and a simultaneous cooling of the surface for about 2-3 months after the injections. The atmospheric heating and surface cooling are mostly pronounced between 30-80$^{\circ}$N, consistent with the latitudinal spread of the smoke plumes. The maximum values for changes in radiative forcing occur within the first 7-10 days after the pyroCb injections, causing a local warming up to 8 W/m$^2$ and a surface cooling up to about 5.5 W/m$^2$.

To put our radiative forcing estimates in perspective, we compare our results with (1) a more recent and much stronger (in terms of injection mass) stratospheric smoke perturbation due to Australian fires of 2019-20 (Khaykin et al. 2020) and (2) our particular case of the BrCo pyroCb but over the ASMA region based on Kloss et al. (2019). The comparisons are tabulated in Table1, along with the respective SSA assumptions and f ratio. Based on Kloss et al. (2019), the f ratio is defined as the ratio between surface and TOA radiative forcing. Both the studies used the UVSPEC (UltraViolet SPECtrum) radiative transfer model to estimate the clear-sky SW surface and TOA radiative forcing perturbations due to the respective stratospheric smoke intrusions. For our model, we compute comparable quantities to the two studies using the differences between our pyroCb and CTL experiments. First, with respect to the Australian fires, our monthly global mean estimates for both TOA (Fig. S2) and surface radiative forcing is smaller by a factor of ~10 compared to Khaykin et al. (2020), which is an interesting finding given the aerosol mass estimates between their study and our differed by a factor of 2. They used OMPS-LP observed stratospheric AOD to derive their forcing estimates.

For the second case, the ASMA aerosol, our mean surface forcing estimates are comparable to Kloss et al. (2019) (their 0.46 W m$^{-2}$ versus our 0.39 W m$^{-2}$), but our TOA forcing estimates are smaller than theirs by a factor of about 2.5. The major reason for the differences in TOA forcing estimates is possibly due to the SSA assumptions between the two approaches. Kloss et al. (2019) assumed a constant SSA range of 0.9-0.93 for the entire SW spectrum in their radiative transfer calculations, but in our model, we have a strong spectral contrast in SSA between 532 nm and 355 nm range for the stratospheric smoke mixture, thus making our model smoke more absorbing, especially in the near-UV range. This is reflected in our respective f ratio values as well. Our study reports a f ratio ~ 4-5.5, while their f ratio is ~2.5. Kloss et al. (2019) also pointed out that a f ratio ~1 is typical of very reflective aerosols (e.g., pollution-sourced aerosols), while a f ratio ~ 3.5 is typical for significantly absorbing aerosol layers (e.g.., biomass burning smoke aerosols). The other reason for differences in forcing estimates could simply be due to the differences in approaches in estimating the perturbations in

radiative forcing. Figures S3 and S4 show our model simulated TOA and surface radiative forcing due to the pyroCb aerosols over the NH. It is evident here that only a small fraction of the ASMA box (in black in Figure S3 and S4) is impacted by the model simulated pyroCb smoke. However, for Kloss et al., only seven SAGE-III retrieved extinction profiles are averaged to represent the extinction/AOD over the entire box/region, and this is further used as an input to their radiative transfer calculations. Therefore, possible under sampling of the region in their study due to limited coverage of SAGE-III observations could have also contributed to the differences in our forcing estimates.

It is worth mentioning here that apart from the aerosol perturbation, dynamical perturbations from the rapid diabatic rise of the heated plume from the tropopause through the lower stratosphere also resulted in enhanced water vapor and ozone depletion in the stratosphere compared to the background state, consistent with the pyroCb plume locations (not shown here). Based on our test simulations, we found that both of these changes resulted in net radiative cooling of the atmosphere and hence provided a negative feedback on the plume rise. This negative feedback is already accounted for in the radiative forcing calculations presented here, since T and Qv blending below the tropopause in our final model set-up (see Section 2.1 for details) allows for these dynamical perturbations to be simulated and the StratChem chemistry module (that includes stratospheric ozone chemistry) is coupled to the radiative transfer scheme within the global model.

### 3.7    Comparisons with Previous Modeling Studies

As briefly discussed in Section 1, our modeling approach to simulate the BrCo pyroCb events is intermediate in complexity between Yu et al. (2019) and Christian et al. (2019) in terms of the treatment of aerosol microphysics and aerosol-radiation and dynamical coupling. Therefore, we compare our assumptions of injection parameters and optical properties for the pyroCb-emitted aerosols with these two previous studies (Table 2) to put our model results into perspective. Table 2 suggests that due to the lack of observations and great deal of uncertainty associated with the aerosol composition (BC to BrC ratio) or absorption properties of the pyroCb plumes, each of the studies optimized their simulations of plume rise by finding a balance between BC amounts, absorption efficiency of individual aerosol components and injection heights. Both Yu et al. (2019) and our study accounted for the impact of aerosol-radiation interactions on aerosol vertical transport via self-lofting, and thus were able to simulate the observed plume rise and hemispherical spread even with relatively lower injection altitudes (close to the tropopause), while Christian et al. (2019) compensated for the lack of aerosol-radiation coupling by injecting the smoke aerosols at higher altitudes (~14 km), which was well within the stratosphere. However, even with the lowest injection altitudes, our model simulations are able to demonstrate the long-term smoke transport pattern in good agreement with the observations, including the transport of the pyroCb smoke plumes over the ASMA region, which neither of the other studies demonstrated.

We further highlight some of the similarities and differences in our major findings. Figure 11 depicts the rise and gradual depletion of the stratospheric smoke mass in our model, suggesting an e-folding time of ~140 days. This is consistent with both the previous studies mentioned above and observations from SAGE-III depicted within Yu et al. (2019). However, Yu et al. (2019) had to implement a photochemical reaction scheme between organics present in the smoke and ozone in the stratosphere to match the observed decay. By contrast, here and in Christian et al. (2019) the smoke lifetime is not mediated by this additional chemistry mechanism and the pyroCb smoke lifetime is simply

the dynamical lifetime of the smoke in the model that includes the removal by large-scale circulation and aerosol sedimentation.

Next, we compare our model estimates of radiative impacts of the pyroCb aerosols on the stratosphere. To this end, we present the SW heating rates calculated by GEOS over September 2017 (Fig. 12a) for direct comparison with Figure 4 of Christian et al. The major differences are in the magnitudes of heating rates (K/day), wherein our estimates

are about a factor of 20-25 lower in magnitudes than estimates of Christian et al. (2019). This is possibly due to the higher amounts of BC (6%, 24 kilotons) injected in Christian et al. (2019) compared to our 2.5% BC (7 kilotons), thereby contributing to the stronger absorption of SW radiation by the pyroCb smoke. Moreover, the horizontal spread of the of smoke plumes also influence the magnitudes of heating rates via aerosol optical depths. Since majority of the plumes in Christian et al. are concentrated toward the high latitudes, higher AODs over this region contribute to

the maximum heating rates concentrated over the poles in their simulations. For our study however, heating rate maxima occur between 40-60$^{\circ}$N, with significant SW heating extending up to the tropics (~20$^{\circ}$N) as well. This can be attributed to the accurate simulations of the transport and subsequently the hemispherical spread of the pyroCb smoke plumes in our study.

## 4    Conclusions

We used the GEOS AGCM to model the emissions and three-dimensional evolution of the smoke aerosols emitted in the extreme pyroCb events that occurred in August 2017 over BrCo. We demonstrated that GEOS is able to simulate the transport, rise, hemispherical spread and lifetime of the pyroCb-emitted aerosols in close agreement with observations from OMPS-LP. We found that aerosol self-lofting plays the most important role in plume rise, and specific to our model, having a constrained large-scale flow via replaying to reanalysis wind fields, was crucial in

closely simulating the observed horizontal and vertical distribution of aerosols. We further used the model to calculate the radiative impacts of the pyroCb-emitted aerosols on the stratosphere and on the overall radiation budget of the Earth. We found that the pyroCb-emitted smoke plumes contribute to an additional warming of the atmosphere by 0.6-1 W/m$^2$ for about 2-3 months after the injections. The heating is mainly located in the stratosphere, coincident with the location of the smoke plumes that contain the strongly absorbing carbonaceous aerosols. The atmospheric

heating led to an increase in SW heating rates by 0.02-0.04 K/day for September 2017. At the surface the smoke aerosol plumes caused a cooling that was comparable in magnitude to atmospheric warming. Our forcing estimates, as well as the heating rates are substantially lower than what is reported in Christian et al. (2019) owing to the differences in assumptions of BC amounts (6% versus 2.5% of the total aerosol mass) and the simulated transport between the two studies. Nonetheless, our clear-sky surface radiative forcing estimates due to pyroCb over the ASMA

region (~0.4 W/m$^2$) are comparable with Kloss et al. (2019). Compared to the much larger perturbations due to Australian fires of 2019-20, both our global mean TOA and surface forcing estimates amount to about 10% of Khaykin et al. (2020). Therefore, potential radiative impacts of multiple pyroCb events in a cumulative sense may not be negligible.

The uncertainties in the assumptions of injection parameters and aerosol optical properties in the models exacerbate the uncertainties in estimates of aerosol direct forcing for the pyroCb smoke. Therefore, measurements characterizing the aerosol composition, size distribution, and absorption properties of smoke plumes emitted from future large wildfire events are necessary and critical for model calibration, such that estimations of the radiative impacts of these stratospheric perturbations using global models can be improved.

**Code Availability.** The GEOS model is available from an externally accessible Subversion Software Repository, whose details are provided at https://gmao.gsfc.nasa.gov/GEOS_systems/geos5_access.php.

**Data Availability.** The GEOS model outputs needed to reproduce the results described in this paper are publicly available for download at data.nasa.gov repository (https://doi.org/10.25966/9fv8-6q78), see complete dataset citation under References section. OMPS LP Version 1.5 data (https://doi.org/10.5067/GZJJYA7L0YW2) are accessible from Goddard Earth Sciences Data and Information Services Center (GES DISC) and SAGE-III/ISS version 5.1 data are accessible at the NASA Atmospheric Sciences Data Center (https://doi.org/10.5067/ISS/SAGEIII/SOLAR_HDF4_L2-V5.1). The Level-1 CALIOP data version 4.10 was also obtained from the NASA Langley Research Center Atmospheric Science Data Center (ASDC-Earthdata).

**Author contributions.** PRC, LDO and SD developed the modeling approach and methodology. GT and OT provided, processed and helped with the interpretation of the observational data. SD and PRC performed the analysis and wrote the manuscript. All authors contributed to the editing of the manuscript.

**Competing interests.** The authors declare that they have no conflict of interest.

**Acknowledgements.** We would like to acknowledge the NASA Earth Science Division and GEOS model developmental efforts at GMAO for their support and the NASA MAP funding under GEOS-CCM project (Program Manager: David Considine). S. Das's research at NASA GSFC was supported by an appointment to the NASA Postdoctoral Program, administered by Universities Space Research Association under contract with NASA. Part of the work is also funded by the NASA contract 80NSSC18K0847. The Computing Resources supporting this work were provided by the NASA High-End Computing (HEC) Program through the NASA Center for Climate Simulation (NCCS) at Goddard Space Flight Center.

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

**Figures**

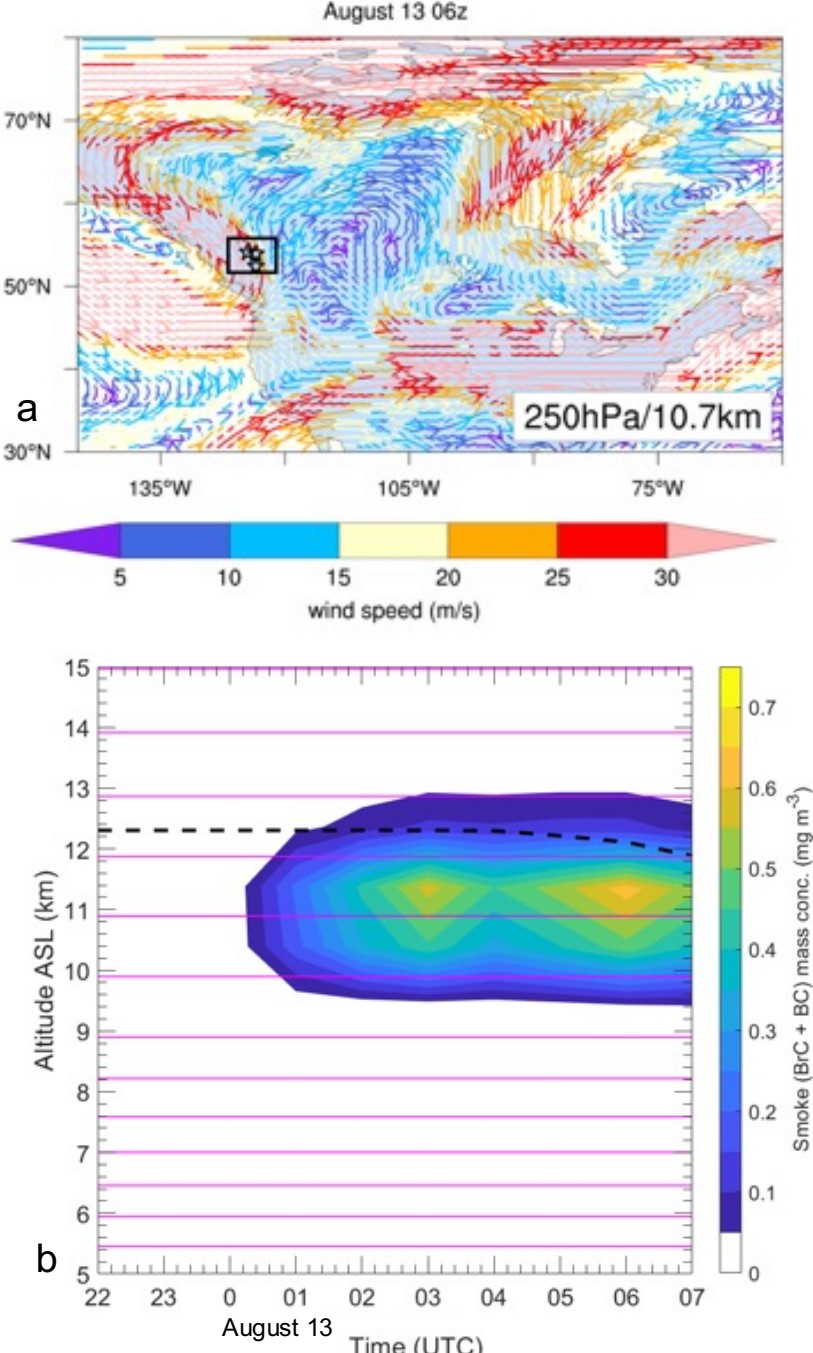

**Figure 1. On the day of initial injections.** (a) Wind streamlines colored by wind speeds on August 13, 2017 at 6 UTC and 250 hPa (or 11 km), which is the mean altitude of smoke aerosol injections in the model. The three black markers depict the injection locations. (b) Vertical location of smoke aerosols is depicted using contours of simulated aerosol (BC + BrC) mass concentrations, averaged over the black box in (a) during the injection period between 0-6 UTC on August 13. The dotted black line depicts the simulated tropopause heights and solid magenta lines depict the edge heights of model vertical levels.

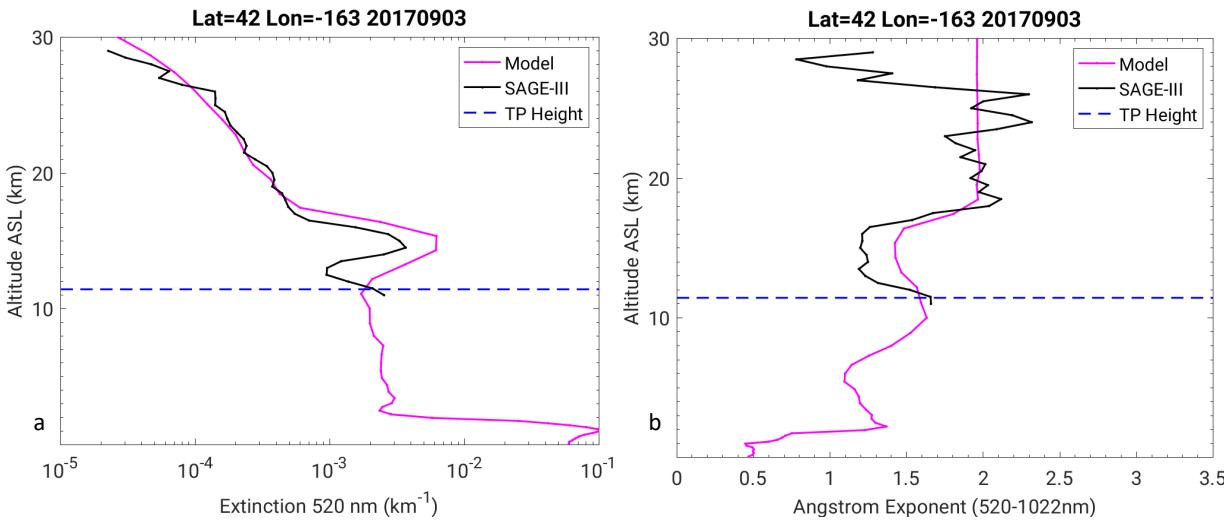

**Figure 2. Calibration of model aerosol size distribution.** (a) Aerosol extinction profiles retrieved from SAGE-III instrument and simulated by the GEOS model at 520 nm for an overpass of SAGE-III/ISS over a pyroCb emitted stratospheric smoke layer at about 14 km on September 3, 2017. (b) The corresponding angstrom exponent (AE) profiles for SAGE-III and GEOS were calculated based on extinctions at 520-1020 nm wavelength pair. Here, the GEOS model assumption of BrC particle size distribution (or PyroCb BrC optics) is calibrated to match the AE obtained from SAGE-III.

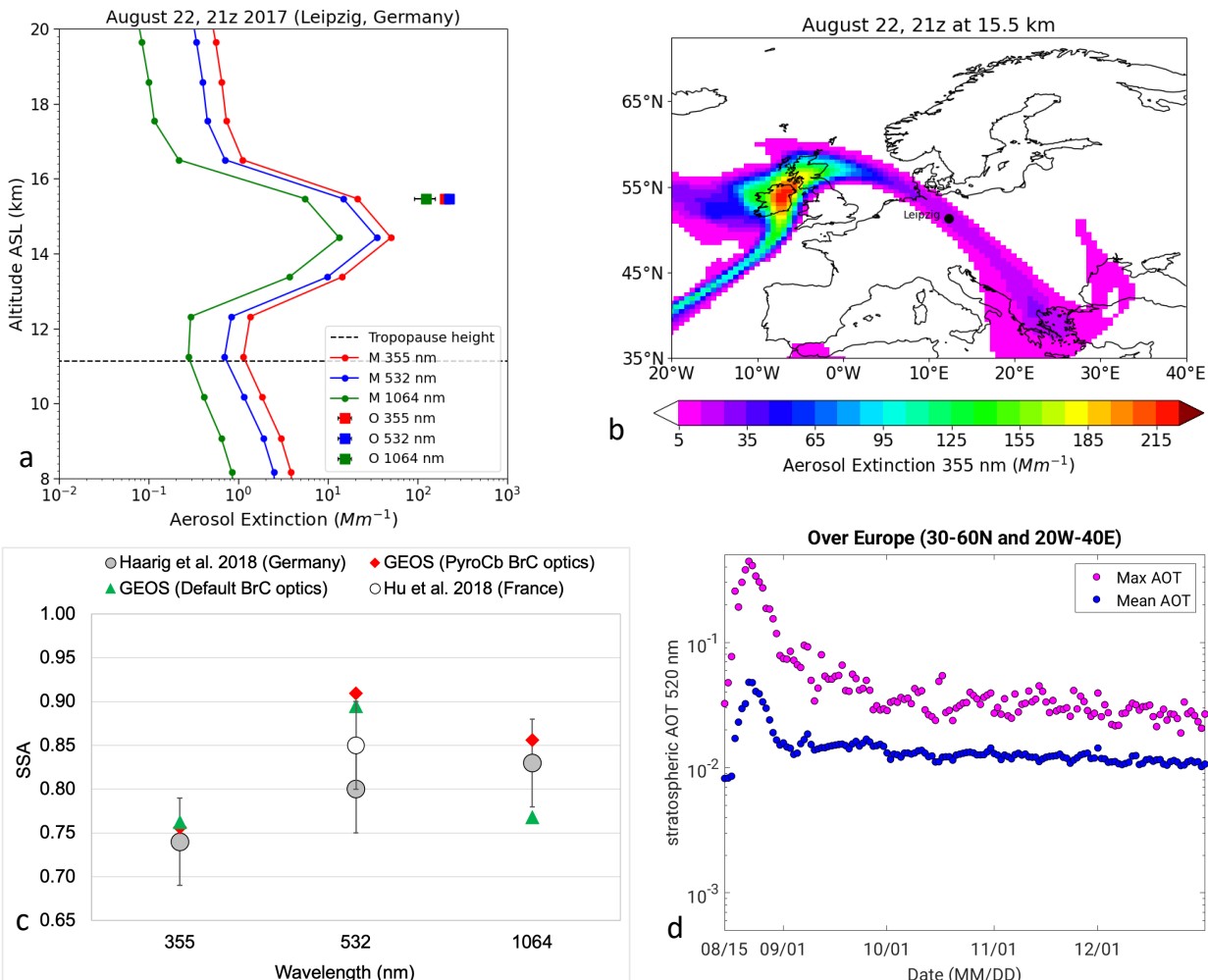

**Figure 3. Evaluation of the simulated aerosol absorption and extinction with ground-based lidars.** (a) Model simulated aerosol extinction profiles (M) and peak extinctions based on lidar observations (O) for the 15-16 km layer reported in Haarig et al. (2018) at Leipzig, Germany on August 22, 2017 at 21z (b) The spatial distribution of simulated aerosol extinctions at 355 nm, around the Leipzig location for the same time. (c) Comparison of single scattering albedo (SSA) retrieved from Raman Lidars (markers with error bars) and simulated by the GEOS model (colored markers) assuming different BrC optics for the stratospheric smoke aerosol layer observed over Europe (Haarig et al. 2018, Hu et al. 2018) after 10-15 days of the pyroCb injections. (d) Evolution of model simulated stratospheric (mean and maximum) aerosol optical thickness (AOT) over Europe from mid-August to end of 2017.

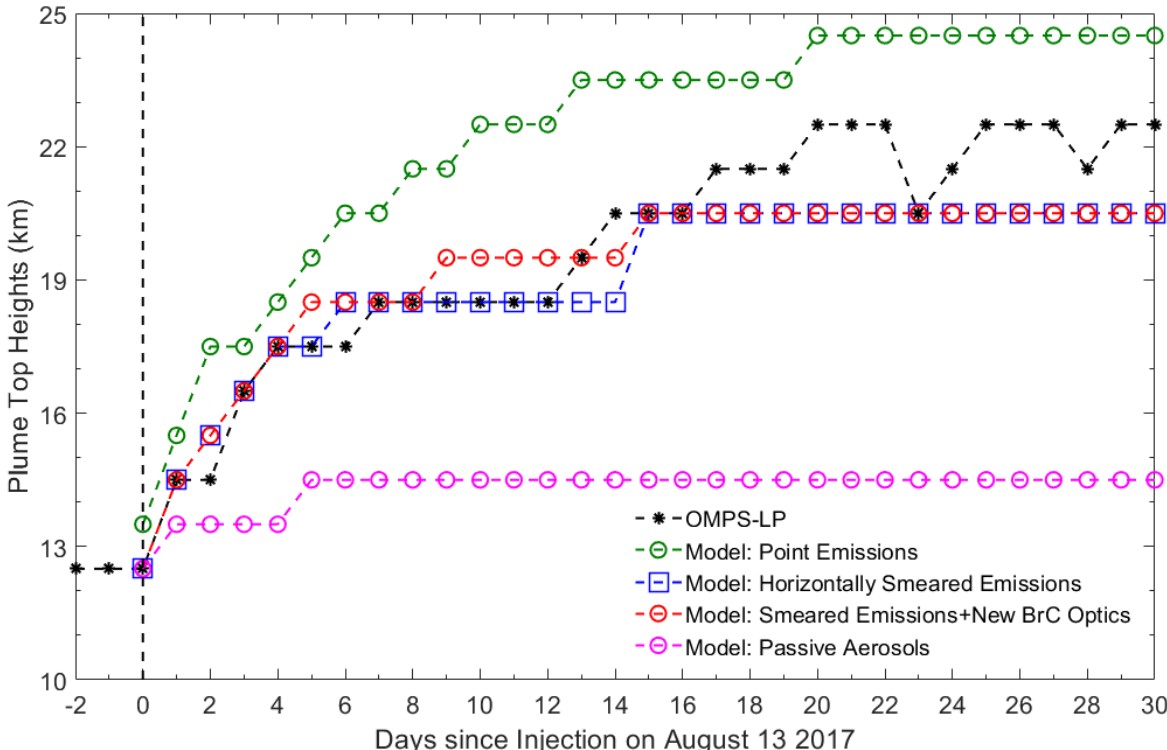

**Figure 4. Sensitivity of plume rise to different assumptions of injection parameters, aerosol optical properties and aerosol-radiation coupling.** OMPS-LP derived plume top heights (km) are depicted in black, while model simulated plume top heights are depicted in colored lines, where each color represents a different model assumption. Plume top heights are defined as the maximum altitude at which mean aerosol extinction (including pyroCb smoke) exceeds the mean background aerosol extinction. The mean aerosol extinction and mean background aerosol extinction were calculated by averaging the zonal means over 30-90°N. See details of plume top height definition in Section 3.2.

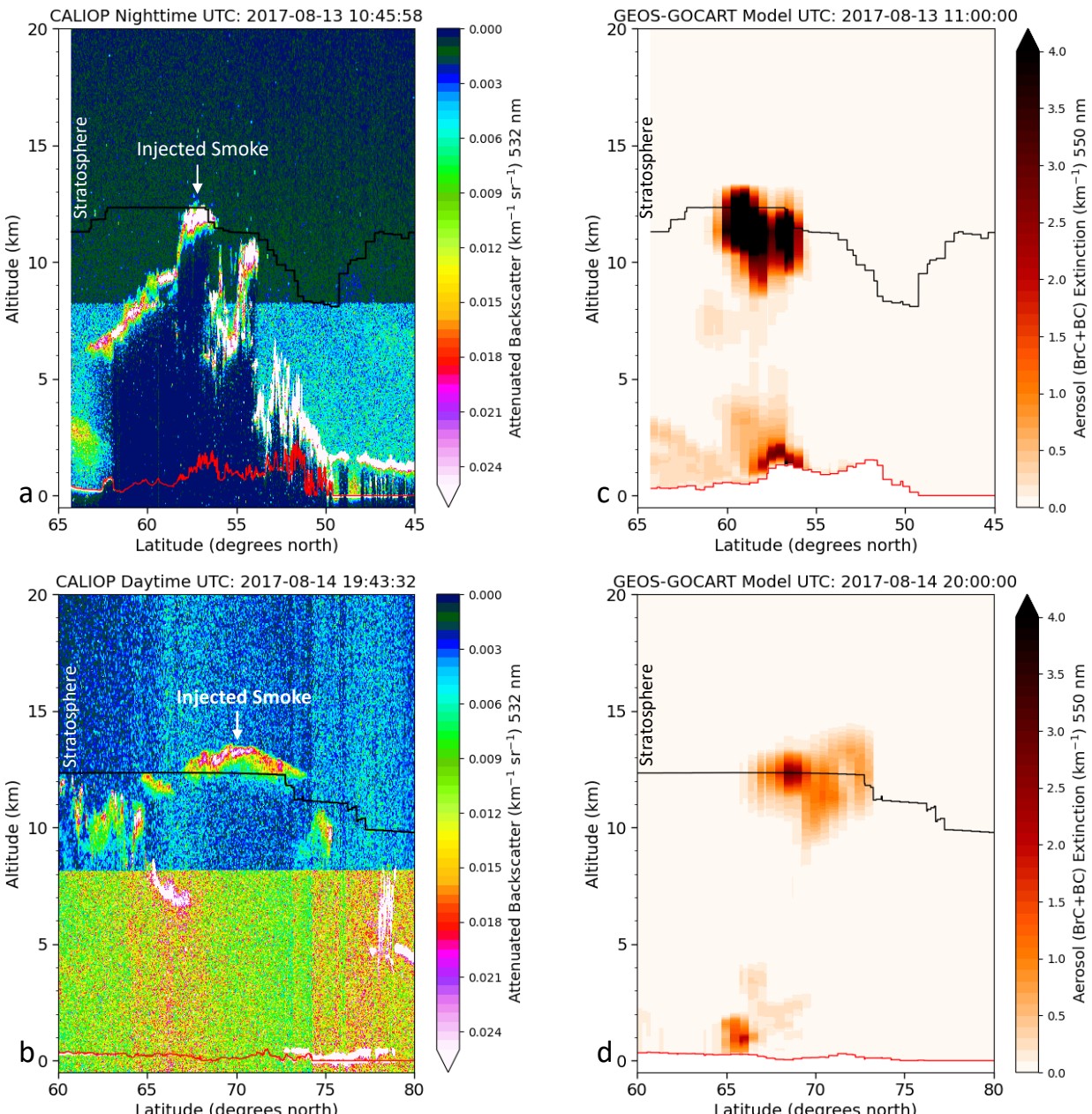

**Figure 5. GEOS comparisons along CALIPSO tracks.** (a), (b) CALIOP retrieved total attenuated backscatter (km$^{-1}$ sr$^{-1}$) profiles and (c), (d) GEOS simulated smoke aerosol (BrC + BC) extinction (km$^{-1}$) profiles along the August 13 ~11 UTC and August 14 ~20 UTC CALIPSO tracks respectively. The red lines depict the surface elevations and the black lines depict the tropopause heights on each panel. Please note that to reduce the downlink data volume, CALIOP Level 1 profile products are reported at different vertical resolution for different altitude regimes. Since the vertical resolution changes around 8.3 km from 30 m to 60 m, a stark gradient is visible in both (a) and (b) for altitudes above and below the 8.3 km level

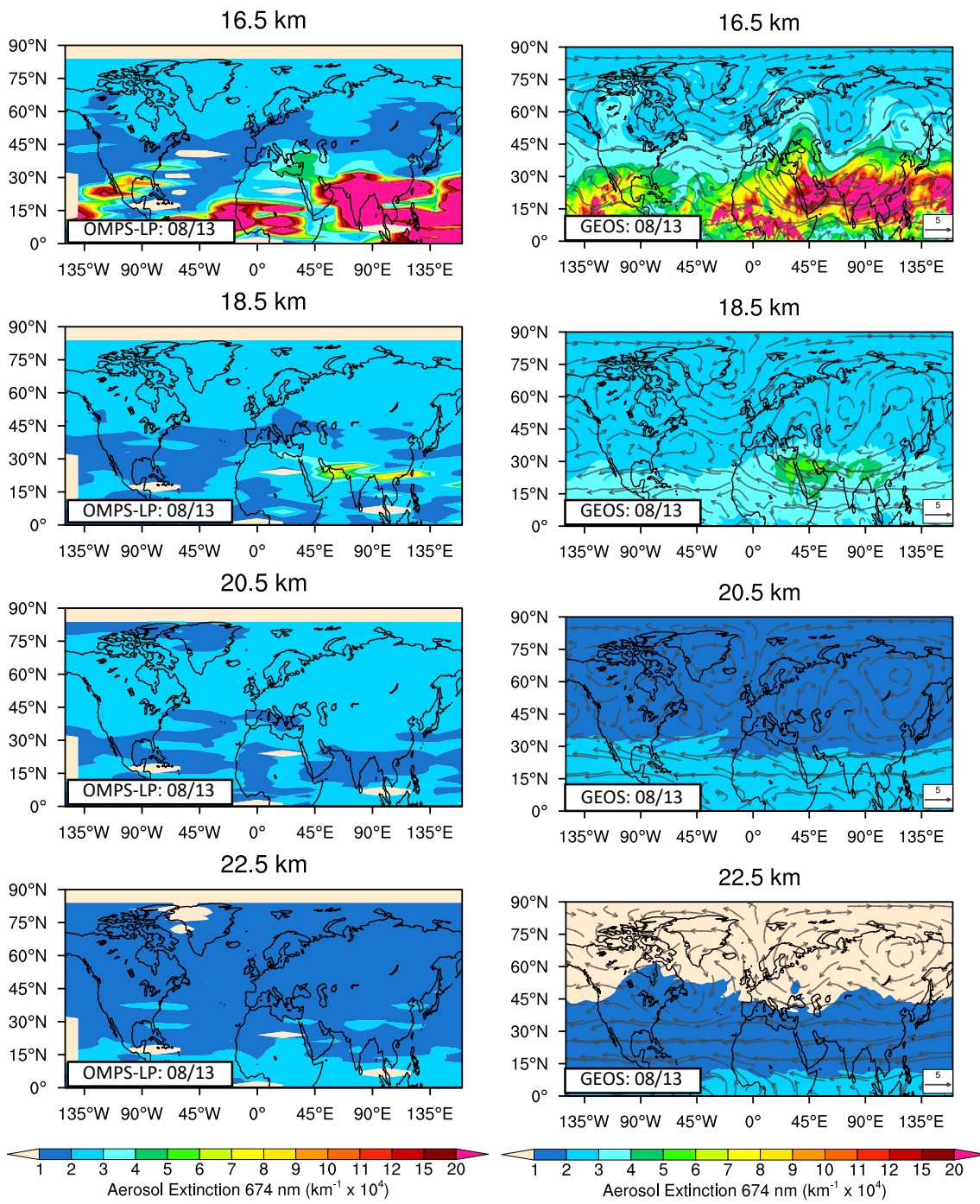

**Figure 6a. Background state on injection day.** OMPS-LP retrieved (left column) and GEOS simulated (right column) total aerosol extinctions (km$^{-1}$ x 10$^4$) at 674 nm at altitudes from 16 to 22 km (top to bottom) on the injection day of August 13, 2017. The simulated wind vectors are overlaid on the model contour plots to depict the trajectory of transport at different altitudes and identification of the anticyclonic flow of Asian Summer Monsoon.

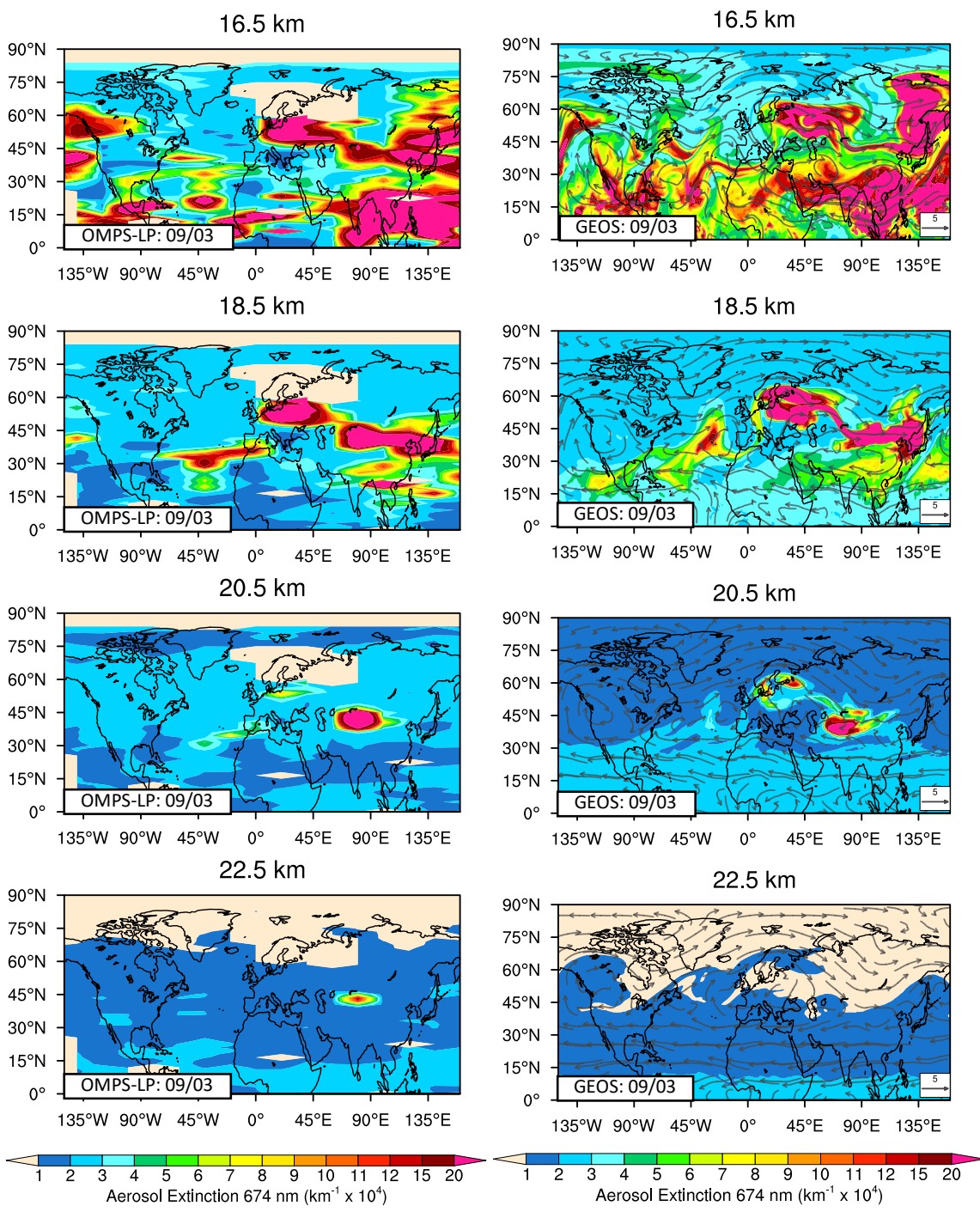

**Figure 6b. Horizontal and vertical transport of smoke plumes**. This is same as Figure 6a, but three weeks after the PyroCb injection. The model wind vectors demonstrate the role of large-scale flow in the horizontal and vertical transport of stratospheric smoke plumes, especially over the ASMA region (15-45°N and 40-110°E).

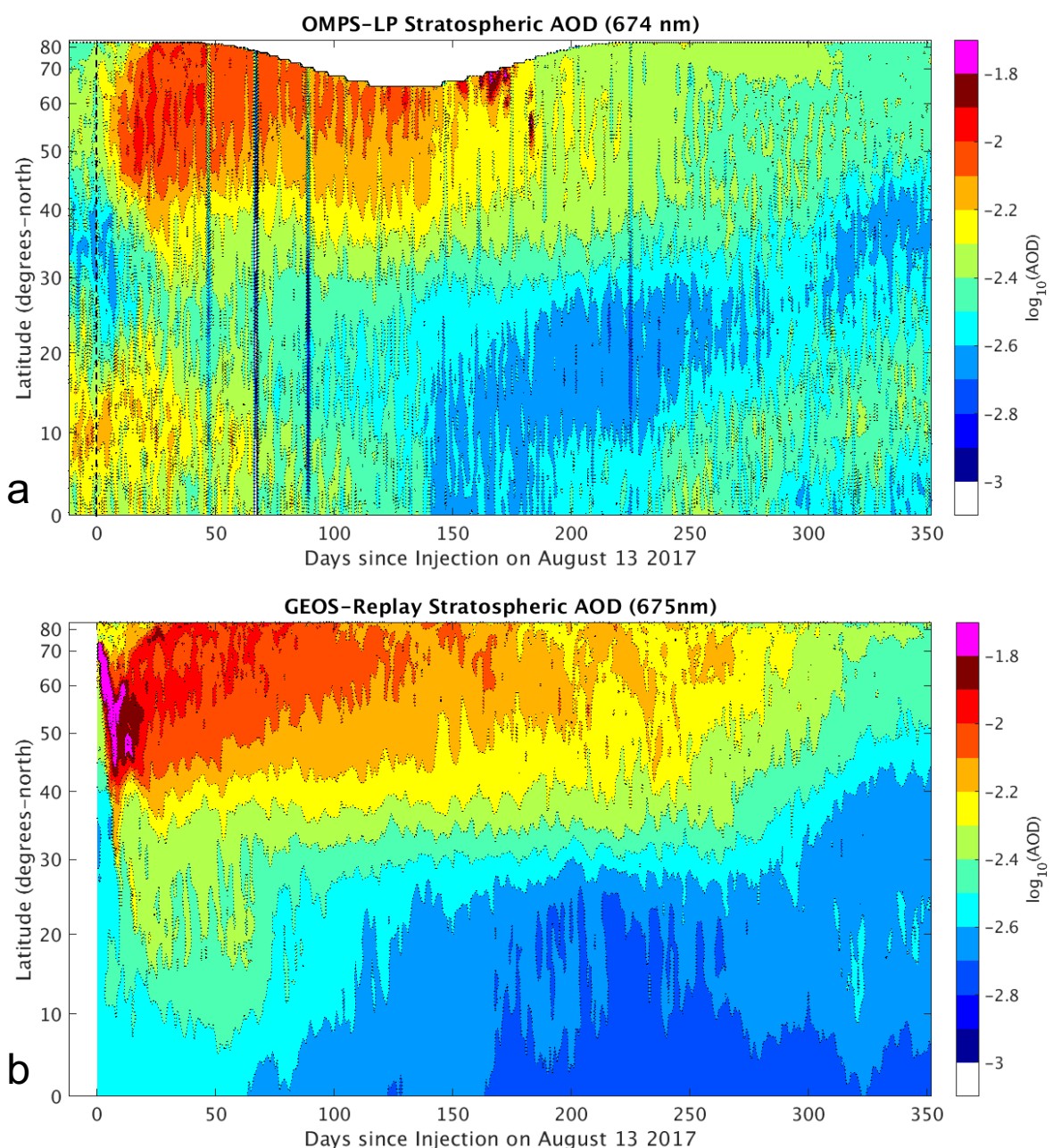

**Figure 7. Hemispherical spread and residence time of aerosols**. Zonal mean of total (smoke + background aerosol) stratospheric AOD (a) retrieved from OMPS-LP and (b) simulated by the GEOS model for about a year after the injection on August 13, 2017. Note that enhanced AODs in the Arctic around day 150 are contribution from polar stratospheric clouds (PSC) due to cloud-unfiltered OMPS-LP data.

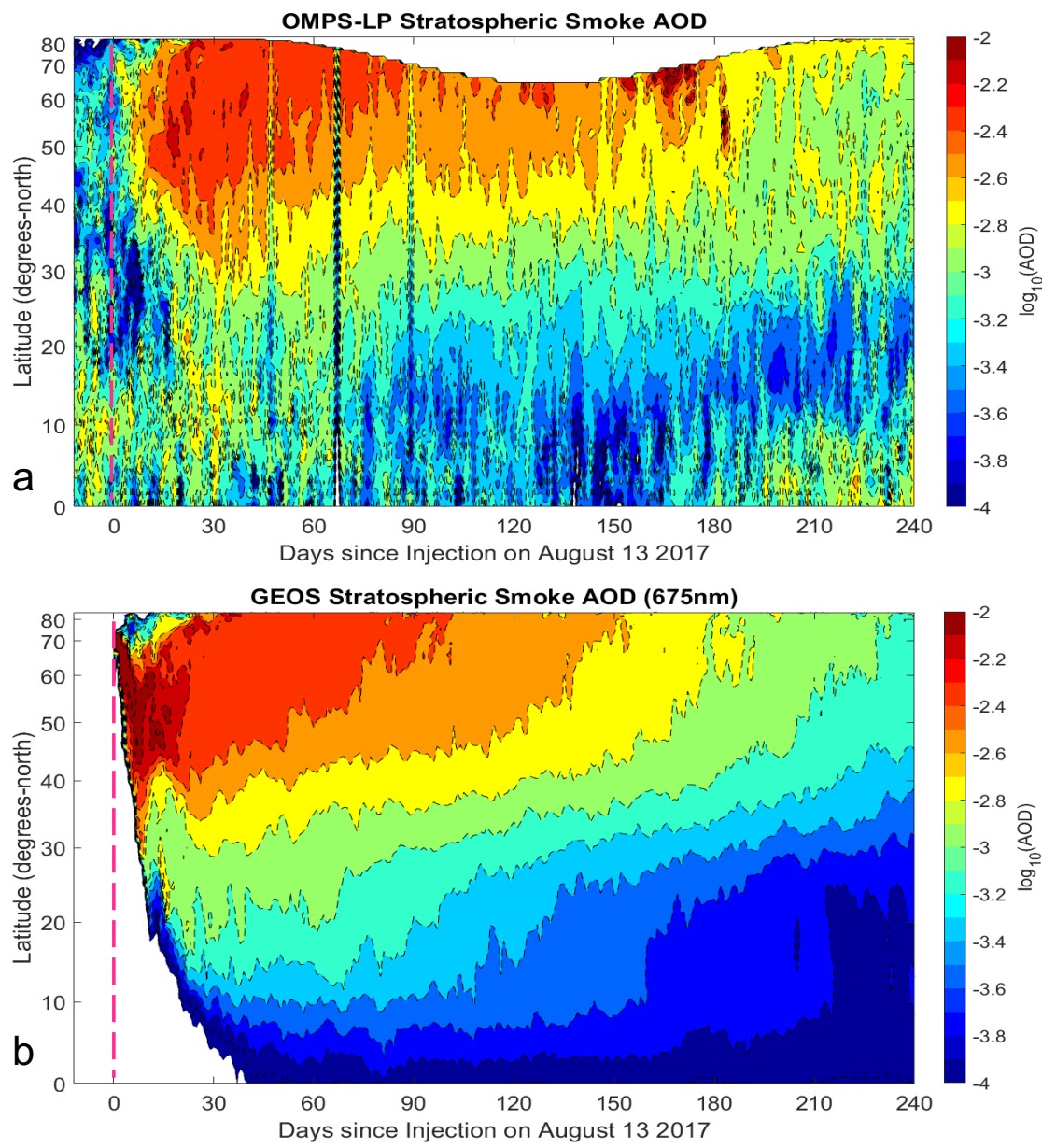


**Figure 8. Hemispherical spread and residence time of PyroCb smoke.** Zonal mean of stratospheric smoke AOD (a) retrieved from OMPS-LP and (b) simulated by the GEOS model over the Northern hemisphere for about eight months after the injection, during which OMPS-LP observed significantly enhanced values of aerosol extinctions compared to the background.


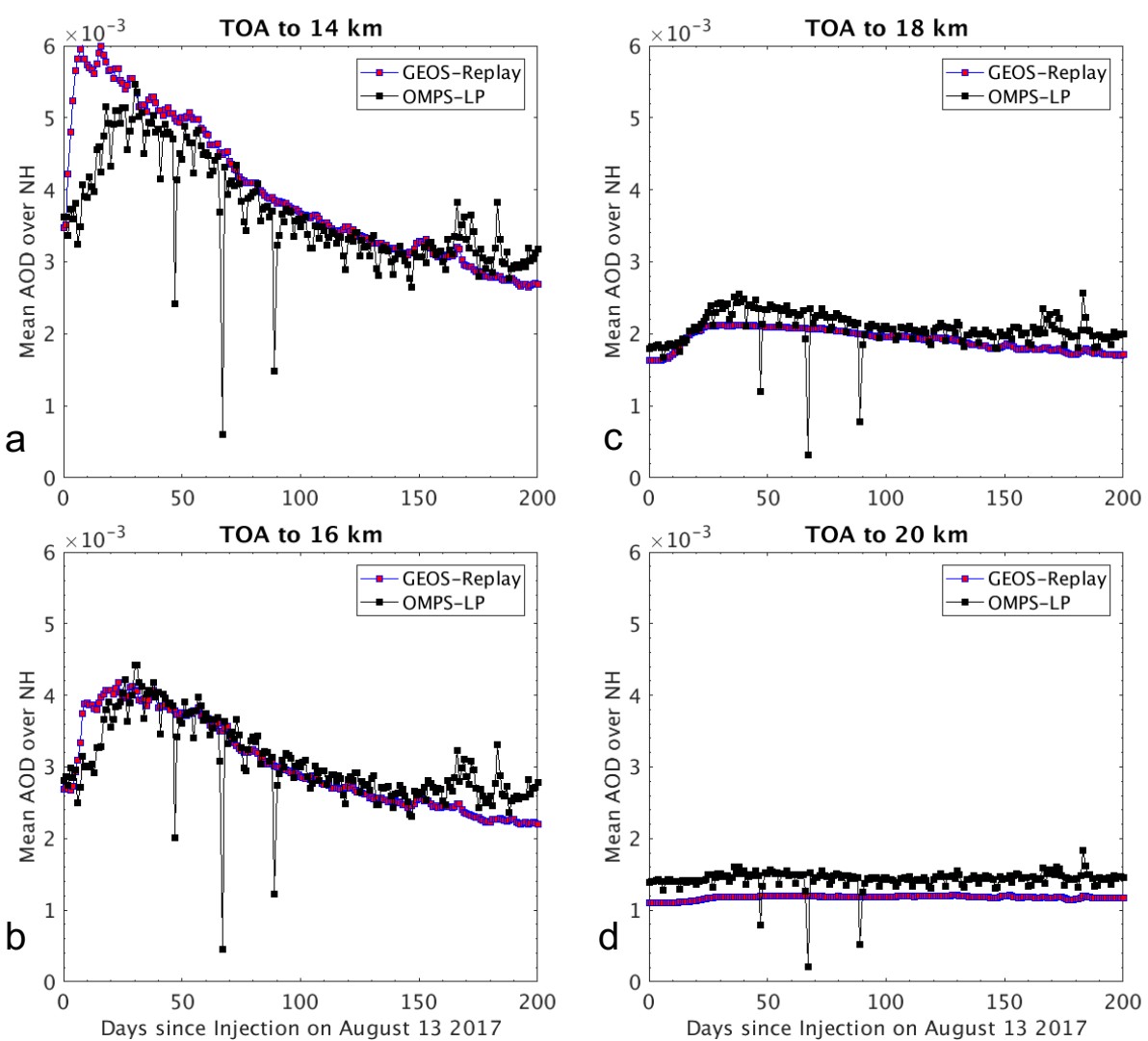

**Figure 9. Vertical distribution of stratospheric aerosols containing PyroCb smoke.** Timeseries of total AOD derived from OMPS-LP (black) and the GEOS model (magenta), averaged over the Northern hemisphere for atmospheric columns extending from top of the atmosphere (TOA) to (a) 14 km, (b) 16 km, (c) 20 km and (d) 22 km.


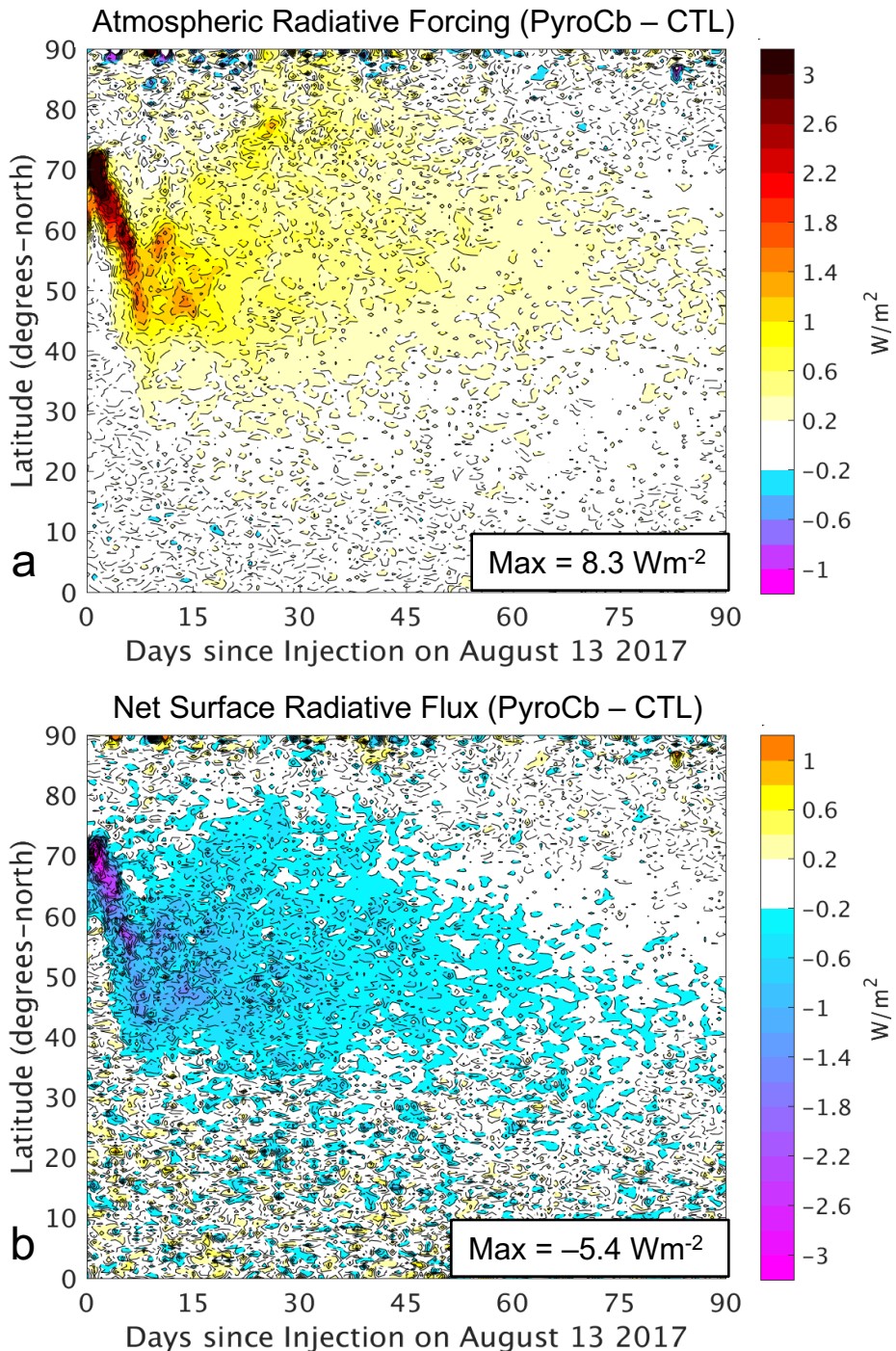

**Figure 10. Impacts of PyroCb emitted aerosols on Radiation balance.** Differences in zonal mean all-sky (a) atmospheric radiative forcing and (b) net surface radiative flux between the PyroCb and Control (CTL) experiments. The maximum values are listed at the bottom corner of each panel since color scales are saturated for high AOD values during the initial days after the injection.

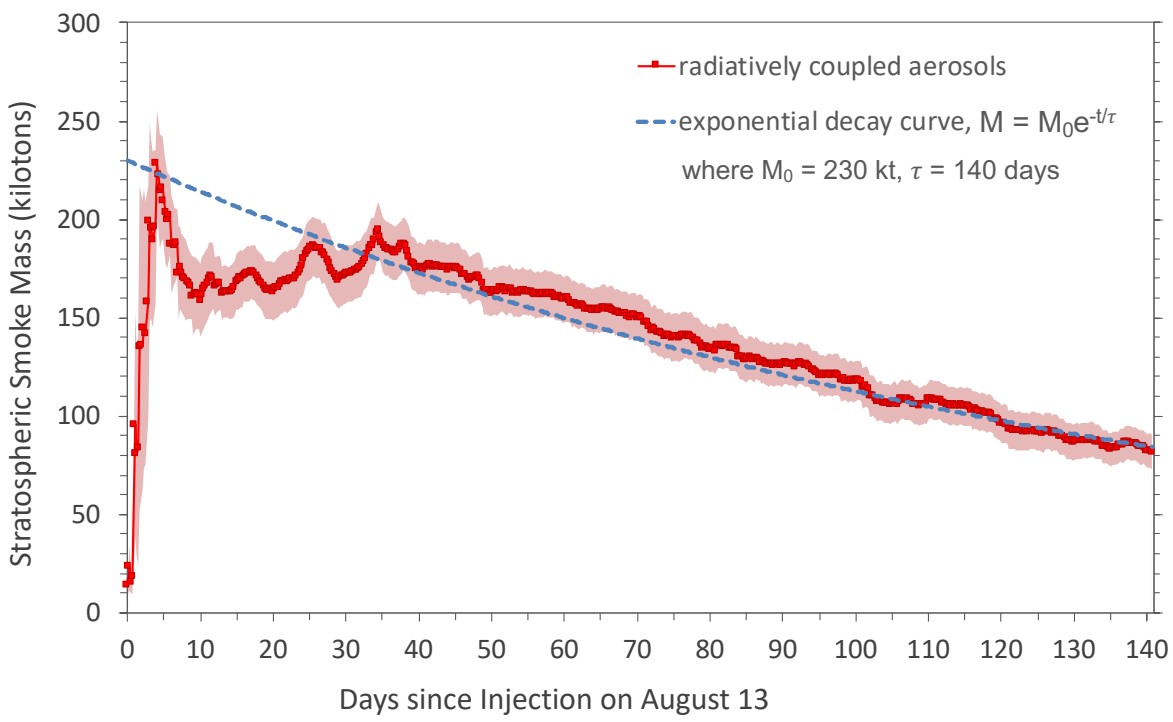

**Figure 11. Decay rate of stratospheric smoke mass.** The variation of model-derived stratospheric smoke (BC + BrC) aerosol
mass (kiloto[...]
inclusion or[...]

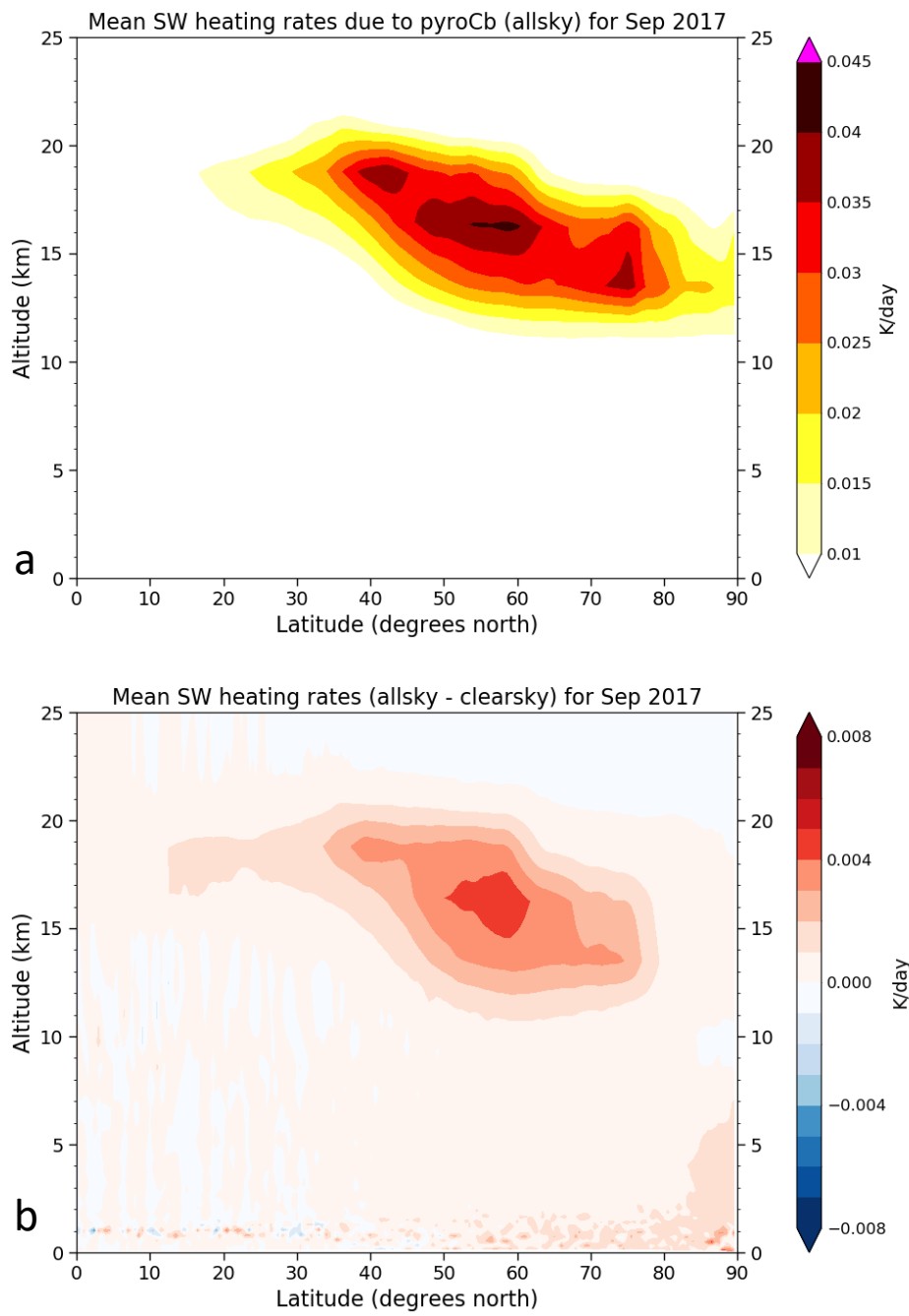

**Figure 12. Heating rates.** Zonally averaged (a) all-sky and (b) all-sky minus clear-sky shortwave (SW) heating rates for September 2017 (K/day) due to the pyroCb aerosols over NH.

**Tables**

**Table 1.** Comparison of clear-sky TOA and surface radiative forcing estimates from recent stratospheric smoke perturbations.

| | Radiative Forcing (clear-sky) | TOA [W m$^{-2}$] | Surface [W m$^{-2}$] | f ratio = Surf/TOA forcing | SSA Assumptions |
|---|---|---|---|---|---|
| **Global mean (area-weighted)** | Australian pyroCb: February 2020 Khaykin et al. (2020) | -0.31 $\pm$ 0.09 | -0.98 $\pm$ 0.17 | 3.2 | Different perturbations with 0.85, 0.9, 0.95 |
| | BrCo pyroCb: September 2017 Our Study | -0.03 $\pm$ 0.01 | -0.12 $\pm$ 0.03 | 4.0 | 0.9 at 532 nm, and 0.75 at 355 nm (Fig. 3c) |
| **Regional mean over extended ASMA region (15$^{o}$N-45$^{o}$N and 40$^{o}$E-110$^{o}$E)** | BrCo pyroCb: Sep 1-5 2017 Kloss et al. (2019) | -0.18 | -0.46 | 2.5[*] | 0.9-0.93 |
| | BrCo pyroCb: Sep 1-5 2017 Our Study | -0.07 $\pm$ 0.01 | -0.39 $\pm$ 0.02 | 5.5 | 0.9 at 532 nm, and 0.75 at 355 nm (Fig. 3c) |

[*]Kloss et al. (2019) report a value of 3.5 in their text and also their Figure 5, but we tabulate the number (2.5) based on their reported mean forcing estimates and the definition of f ratio.

**Table 2.** Assumptions of injection parameters and optical properties for the pyroCb-emitted aerosols in different modeling
studies.

| | Aerosol Injection Parameters | | | | Optical Properties (550 nm) | | |
|---|---|---|---|---|---|---|---|
| Study/Paper | Total (Tg) | BC (%) | Other (%) | Heights (ASL) | BC Refractive Index | OC/BrC Refractive Index | Mixing State |
| **Our Study** | 0.3 | 2.5 | 97.5, BrC | 10-12 km | 1.75 - 0.45i | 1.47 - 0.016i | External |
| **Yu et al. (2019)** | 0.3 | 2 | 98, Organics | 12-13 km | 1.95 - 0.79i | 1.4 - 0.0i | Internal for BC aggregates coated with OC, otherwise external |
| **Christian et al. (2019)** | 0.4 | 6 | 94, OC | 13.7 km (0.2 Tg) + 0.2 Tg between surface and 13.7 | 1.75 - 0.45i | 1.53 - 0.006i | External |

