# Peer review of "The Long-term Transport and Radiative Impacts of the 2017 British Columbia Pyrocumulonimbus Smoke Aerosols in the Stratosphere"

_Atmospheric Chemistry and Physics, 2020_

## Referee Comment (RC1) · Anonymous Referee #1 · 19 Feb 2021

I enjoyed reading the paper and believe it should be accepted for publication after the consideration of a few comments.

The presented manuscript is a nice comprehensive overview modelling study of the dispersion and radiative impact of the British Columbia stratospheric fire plume from 2017. GEOS simulations represent the observed transport (vertically and horizontally) including the self-rising feature of the plume and aerosol properties in form of aerosol extinction well, which is of big asset for the investigation of fire plume transport mechanisms.

Confusions especially about the Asian summer monsoon dynamics should be revised for the new version of the manuscript. Have a look at my comments below. A lot of them are only ideas and suggestions. At some places I would have wished for more clarifications and comparisons with other studies (especially your radiative forcing results).

**Comments:**

**line 344 to the end of the page** does not seem correct/logical at this point. You want to justify why OMPS and GEOS see enhanced aerosol extinction values at those 'high' altitudes, which is not so much related to the BDC. This (or similar) is what you could write: During the Asian summer monsoon tropospheric trace gases and aerosols are convectively lifted into the UTLS, where they remain largely confined within the transport barriers of the Asian monsoon anticyclone during the monsoon season (June-September) (references). During August 2017 the center of the ASMA was between 15-45N and 40-110E as defined in Kloss et al., 2019. This explains…

For the references here: I would not use Randel et al., 2010: They quickly mention the tracer enhancement within the ASMA, but the focus of this paper is about what happens after the break down of the ASMA (September/October), i.e. where do those 'isolated' air masses go. And even this mechanism has been extensively discussed and revised since then. Only a very small fraction will end up in the ascending branch of the BDC, which would not show in your data. I would use Park et al., 2008 (as an early paper) https://acp.copernicus.org/articles/8/757/2008/ or/and Santee et al., 2016 (a comprehensive more recent study) https://agupubs.onlinelibrary.wiley.com/doi/full/10.1002/2016JD026408 Both those studies focus on the trace gas enhancement within the ASMA.

Rather than citing Vernier 2015, I would take the original Vernier 2011 paper (where the ATAL: aerosol enhancement within the ASMA was first discovered): https://agupubs.onlinelibrary.wiley.com/doi/full/10.1029/2010GL046614

AND for your specific case in 2017 have a look at Kloss et al., 2019. They show the ATAL signal in 2017 in mid-August up to 18 km altitude with SAGE III data.

**Sentence starting in line 354:** No, this is not the ascending branch of the BDC, but rather a result of the circulation around/above the ASMA. Please have a look and refer to Wu et al., 2010: https://acp.copernicus.org/articles/17/13439/2017/

And Tissier and Legras, 2016: https://acp.copernicus.org/articles/16/3383/2016/

This feature is also seen in Vernier et al., 2011 (have a look at their Figure 3 e.g.)

And again have a look at and refer to Kloss et al., 2019. There you even see for your specific example that the fire plume is around 2-3 km higher than the ATAL (their Figure 2). Also, mean SAGE III profiles at roughly the same time/region seem 1-2 km lower in altitude than what you observe with OMPS. The SAGE III profile matches the model results better. Maybe this might be worth mentioning?

**Sentence starting at 373:** This is a feature often seen after volcanic eruptions as well. You could cite Haywood et al. 2010 (https://agupubs.onlinelibrary.wiley.com/doi/full/10.1029/2010JD014447) for OSIRIS-HadGEM2 comparison after the Sarychev eruption (their Fig. 5) and Kloss et al., 2021 (https://acp.copernicus.org/articles/21/535/2021/) showing the same feature with OMPS data (compared to WACCM) for the Raikoke eruption (2019).

**Section 3.6:** Please compare your radiative forcing values with the ones from Kloss et al., 2019 (UV Spec simulation based on SAGE III data in the ASMA region), they seem to match well. Furthermore (see also line 480), instead of or additionally to the Pinatubo comparison it might be worth bringing the radiative forcing estimations of your study from the BC fires in context to the most recent (much stronger) Australian fires. Please have a look at Khaykin et al., 2020 (https://www.nature.com/articles/s43247-020-00022-5). If you'd like to compare your radiative forcing results to all stratospheric aerosol events of the past few years, also have a look at Kloss et al., 2021 (Figure 8 and 9 for radiative forcing estimations of the Raikoke plume).

**Section 3.7 and general**: The model from Yu et al., 2019 was not nudged after 12 August 2017 (see their supplements), which limits the trustworthiness of the early distribution. Might this be worth considering when comparing your models and to explain some differences?

**Figure 4:** Could you clarify if those lines are averages over the whole NH (as for the background aerosol extinction)? But if so: Even though you only consider data above the cloud top height for OMPS, would the result not be biased by different tropopause altitudes (tropics vs. higher latitudes)? How can you distinguish between 'plume clouds' and other clouds for the OMPS observations?

**Minor comments/suggestions:**

Line 1 Erasing the term 'Pyrocumulonimbus events' would make the title more accessible and the pyroCb event itself is also not really the topic of your manuscript. All together your paper is a modeling study with satellite comparisons, I would add the term 'model study' or 'GEOS' in the title. Just a suggestion..

Line 19-21: 'The model simulated… are in close agreement' ?

Line 50: You could include and generally also have a look at Lestrelin et al., in ACPD : https://acp.copernicus.org/preprints/acp-2020-1201/

Around line 51: You might want to add ground based observations as well, e.g. Ansmann et al., 2018: https://acp.copernicus.org/articles/18/11831/2018/ Baars et al., 2019, Khaykin et al., 2018..

A lot of the literature within the paper is not found in the reference list: e.g. Peterson et al., 2016; De Laat et al., 2012; Chen et al., 2016; Chen et al. 2020 …

Line 91/93-100: Don't you think the model description is a bit too detailed for the introduction? You might consider shifting it to the methods section.

Line 161: The 'baseline experiment includes the injection'….?

Section 2.2: The newest OMPS version is 2.0, giving aerosol extinction observations on multiple wavelengths. I am wondering, why you didn't use the latest version.

Section 2.3: Why did you choose to bring SAGE III and CALIOP in one methods section?

Section 2.3, SAGE III: For OMPS you explain how you treat clouds, how about SAGE III? I understand that you only show one profile and the AE shows very obviously that there were no clouds at that point, but you also write that this is only an example.

Throughout the manuscript you change between 'SAGE III', 'SAGE-III' and 'SAGE-III/ISS'. I think the last version is the best..

Line 239: 42°N seems too north to be called 'sub-tropical'

Line 245: Maybe you could add a '(not shown here)'

Line 323-328: Nice comparison. Maybe you could state the motivation of showing this Figure. In my eyes, this paragraph seems a bit lost at the beginning of 3.3.

Line 338: nor -> neither?

Line 391: 'It is clear from the comparison,…, (that?) the magnitudes…'

Line 397: You mean 'down to' I assume..

Line 412: compared

Line 442: I thought you wrote earlier (section 3.3 / Figure 4) that the lofting around the ASMA is not well represented in the model.

Line 466: August 2017 over..

Line 477: that-> than

In Data Availability: You might include the data version for SAGE III and CALIOP

Figure 1: Could you replace the titles by more meaningful ones?

Figure 6a: km-1

---

## Short Comment (SC1) · 21 Feb 2021

This paper will become an important contribution to the stratospheric smoke literature! That motivated me to write this comment.

Baars et al. (ACP, 2019) presented a dense set of lidar network information on geo-metrical, optical and microphysical properties of the stratospheric smoke over Europe after the strong pyro-CB-related smoke event of August 2017. You mention the paper briefly in your article. The paper covers six months of smoke observations!

The Baars et al paper should be mentioned already in the introduction as it is an im-portant observational contribution to the research and documentation of the record-breaking smoke event, that you are modelling.

Furthermore, the European lidar network results should then be compared with your model findings (for Europe).

I am curious to see how your model results agree with this height-resolved smoke lidar data set!

How well do the model results agree with the lidar data in terms of optical depth or even layer-mean extinction coefficient?

Does the model resolve properly the height range of smoke observed over Europe, from Northern Norway to southern Portugal and Spain (western Mediterranean) and Cyprus and Israel, in the Eastern Mediterranean.

To be more precise:

Figure 2: Why did you not use the Baars-et-al.-2019 data (although knowing this paper and the results) in the comparisons shown here?

Figure 3: Here, you use lidar data from Europe (even from Leipzig) ! Very good, thank you!

Figure 4: Here, it would make really sense to take the European lidar data (on smoke layer top heights) to check the quality of the model results.

Figure 6c, 6d, 6e: another excellent opportunity for lidar (Baars et al.) vs model com-parisons, ...with Europe in the center of all your plots.

Finally, Figure 7 and Figure 8 results should or could be compared with the extinction coefficients presented by Baars et al. (2019).

---

## Referee Comment (RC2) · Anonymous Referee #2 · 23 Feb 2021

**Review of Das et al. "Pyrocumulonimbus Events over British Columbia in 2017: The Long-term Transport and Radiative Impacts of Smoke Aerosols in the Stratosphere".**

The manuscript by Sampa Das and Coauthors addresses a topical issue of the physical properties and behavior of PyroCb smoke plumes in the stratosphere by simulating their spatiotemporal evolution using GEOS general circulation model with prognostic aerosol module. Compared to previous modeling studies of BC smoke in the stratosphere, Das et al. perform a careful estimation of the smoke optical properties via constraining the simulations of smoke evolution by available observations. In view of the emerging realization of the effects of extreme wildfires on the stratosphere, the accurate information on the radiative properties of stratospheric smoke, in particular the SSA, is certainly valuable. The paper is well organized and fluently written, the conclusions are substantiated by the presented material, which altogether renders the paper suitable for ACP. There are however several issues that require important revisions.

My major concern is that the authors take no account for the underlying clouds that could significantly enhance the heating of stratospheric smoke (e.g. Boers et al., 2010). This issue is not even mentioned in the paper, which is surprising given the effort the authors invest to accurately constrain the smoke optics. This shortcoming should lead to underestimation of the diabatic ascent rate, which actually seems to be the case judging from Fig. 4, where the observed plume top appears 2 km higher than the simulation after T+20 days. Obviously, a global-scale simulation including the clouds would substantially complicate the modeling experiment. However, it would be of great value for this study if the authors estimate how much a convective cloud extending up to the tropopause layer (which were widely encountered in the ASMA region during that time) would affect the heating of the stratospheric smoke plume.

**Specific remarks**

Figure 2. It would be interesting to see the simulated and observed profiles in the same plot

Figure 3a. Would it be possible to display the peak values from Leipzig observation?

Figure 6. The amount of information conveyed by this figure does not justify 40 panels. I believe most of them could be moved to supplementary material. I also have a concern on how the OMPS-LP data are presented in this figure: there appears to be a strong boxcar smoothing in the zonal direction, which was applied to mask the gaps between the orbits. It might be a better solution to aggregate OMPS-LP data over 3 days to avoid smoothing across these gaps.

Figure 7. The enhanced SAOD in the Arctic after day 150 is obviously due to PSCs. This should be mentioned in the text and/or figure caption. Alternatively, the PSCs could be removed using the same approach as in OMPS-LP V2.0 retrieval, which would enable a better comparison with the simulation.

p.10, l. 340. "… the model overestimates the background aerosol extinctions…". First, it is not totally obvious from Fig 6a that the model actually shows significantly higher extinctions. Secondly, the strongly enhanced extinction in the tropics is most certainly due to TTL cirrus and not due to aerosols.

p.10, l. 344 – 348. This sentence contains several statements that are either overly general or not entirely correct. The references provided are not quite relevant too. I would suggest to omit this sentence.

P.11, l. 355 – 365. I disagree with the interpretation provided. The BC smoke plumes were mostly contained in 3 bubbles confined by smoke-charged vortices (SCV), of which only one (Vortex A in Lesterlin

et al., 2020 https://acp.copernicus.org/preprints/acp-2020-1201/acp-2020-1201.pdf) ascended to 23-24 km, whereas the two others (B1 and B2) ones did not rise as high. The vortex A was already at 19 km while overpassing Europe (Khaykin et al., GRL , 2018) and by the time it arrived to the Asian region, it was already at 21 km, i.e. well above the Asian anticyclone (Lesterlin et al., 2020;  Bourassa et al., JRG, 2019). Hence, the AMA could not have played much role in terms of the upward transport. Likewise, the cloud scavenging is of no relevance here as the A bubble is well above the TTL clouds. What could be the actual reason why the model falls short reproducing the diabatic rise is the absence of clouds in the simulation.

p.11, l. 375 – 377. Given the large extent of the initial cloud, it is unlikely that it was located in between the 3-slit swaths. I believe, a more important reason why OMPS-LP doesn't get the early cloud is the saturation at extreme extinctions.

p.11, l. 378. Same issue here. The injected smoke cloud is already above the tropopause on 14 August (Fig, 5b) therefore since the cloud screening was applied for clouds below the tropopause (Sect. 2.2), it should not have discarded this plume.

p. 11, l. 381 and p.12, l.392. Total AOD would imply the entire column. Do you mean stratospheric AOD here?

p.12, l.387. rather be said "…there were no strong volcanic eruptions or PyroCb events…"

p.12, l.404. The sentence needs revision, cf. previous remarks.

p.13, l.423. It would be very useful to provide the value of global-equivalent monthly-mean RF to be compared with that of Australian bushfires (Khaykin et al., 2020).

p.13, l.425 – 427. The gaseous composition of the stratospheric PyroCb plumes is vastly different from the background values (strongly enhanced $H_2O$ and CO, depleted $O_3$), which may potentially have an important impact on the plume heating. This should be discussed in a more careful way, i.e. how the alteration of these gases could affect the plume rise.

**Technical remarks**

p.4, l. 114, In August

p.6, l.196. The reference to Chen et al. is missing in the bibliography.

---

## Referee Comment (RC3) · Anonymous Referee #3 · 2 Mar 2021

The quiescent state of stratospheric aerosols formed from precursors welling up from the troposphere is known to be often perturbed by injections from volcanic emissions. In the early years of this century, a completely different source of stratospheric aerosols was discovered in the form of fire-driven thunderstorms or pyrocumulonimbus (pyroCb) events. These events have occurred at mid latitudes in both hemispheres mostly from Canadian and Australian fires and there is a strong focus on these events at this time. This paper is a modelling study of the intense pyroCb events that took place in August 2017 over British Columbia, Canada. There have been a number of modelling

studies of this particular event in recent years attempting to simulate the stratospheric perturbations. Several space borne sensors as well as ground based instruments have provided measurements on this event. The authors calibrated several of the input parameters for their simulations using data from these measurements but more may be necessary for improved modelling. The paper is well within the scope of ACP and I recommend publication after revision.

Major comment:

1. The authors have optimized their model for aerosol size by using the Angstrom Exponent (AE) derived from SAGE III retrievals, but I didn't find any explicit discussion of the possible impact of morphology or shape of the aerosol particles. In particular, one of the most intriguing observations from the Canadian pyroCb events in August 2017 as well as from the Australian events of January 2020 is the high depolarization ratio in the plumes implying significantly non-spherical particles. As shown by Christian et al. (2020), the depolarization ratio for the Canadian pyroCb kept increasing for several weeks after injection. Several explanations have been put forward in the recent papers but the issue is not completely settled, in my opinion. In any case, I believe realistic modelling efforts should address this aspect and its impact on the evolution of the plume. Yu et al. (2019) had specifically addressed this by using fractal aggregates of black carbon. They found that these fractal aggregates produce higher absorption in the mid-visible. I am wondering if inclusion of non-spherical particles will help solve the lower amount of lifting produced by the model in the current study as compared to the observations. I believe the paper would improve significantly by addressing this issue.

Minor comments:

1. Lines 212-213: CALIOP measures both the parallel and perpendicular component of the backscattering signal at 532 nm but only the total backscatter at 1064 nm.

2. Please use the same scale on the x axis for Figures 2a and 2b. In this figure the comparison between model and SAGE III extinction profiles shows a peak at ∼14 km

on September 3. Is this from another episode of pyroCb, since going by Figure 4, the plumes should be near 22 km by September 3? It is not very clear to me if the simulated AE shown here was actually used or is it just to show the methodology used as an example, as the authors state, page 7, line 236? Would it not be better to present this comparison for some cases in August? Why not show only the SAGE III extinction profile for the wavelength being compared at?

3. Since the simulations run for several months for the aging smoke, I wonder if a time dependent AE and SSA were considered for these simulations.

4. The authors present a very brief comparison of the CALIPSO data and simulation in Figure 5. Firstly, why is the color bar for the CALIPSO backscatter placed upside down? It can be a little confusing. Also, why is there an apparent discontinuity near 8 km in both the panels? The scene shown in Figure 5a was also shown in Torres et al. (2020) but using the directly available CALIPSO browse image and this discontinuity was not seen in that figure. I suspect it is a plotting artifact and related to the vertical resolution of the downlinked data changing around this altitude. This should be clarified in the text for the benefit of readers. This plume observed by CALIPSO actually has interesting constraints for modelling. As mentioned in Torres et al. (2020), this plume (Figure 5a) is a mixture of ice and smoke, with rather high depolarization ratio of 0.2-0.5. While ice is not likely to survive in the stratosphere for too long, this is an ice-smoke mixture and it will be interesting to study if this mixture impacts the evolution of the plume in the first few days when the plume is rising steeply. In fact for the Australian pyroCb events of January 2020, Khaykin et al. (2020) have presented evidence of ice up to 22 km from MLS ice-water content retrievals. Also, I notice that the model missed out those parts of the plume which are more aerosol rich, i.e. the extended plume between 6-10 km between 60N-65N. This would have been clear if the authors had presented the attenuated color ratio image from CALIPSO.

5. I wonder why the authors did not use the latest version 2.0 OMPS data. Perhaps some of the differences between the model and OMPS data noted by the authors could

be alleviated by the new data.

6. Section 3.3 on the link to the ASMA is not particularly convincing. For instance, referring to Figure 6a, I am not sure if I understand the statement on line 347-348. The enhanced values of extinction at 16 km and even at 18 km over south Asia seem to be simply the ATAL feature, which the authors don't seem to mention. Also in addition to these lat-lon plots, a height latitude plot of the extinction might be useful. The legend for Figure 6a may need re-wording, since no smoke is seen in either model or OMPS data in this figure. Similarly the statements in lines 355-360 are not clear or convincing.

7. Figure 9—I am curious as to why there is a bump in AOD around day 180 after the pyroCb injection with the model showing a significant underestimate particularly at 16 km.

8. In Figure 11a, why is there a sharp drop from the peak value to about day 10 and then a slow rise to about day 30 before the steady decline?

9. Lines 454-455—since the 2 estimates differ by such a lot (factor of 20-25), is there any way to evaluate this?

---

## Author Comment (AC1) · 27 Apr 2021

**Response to Referee Comments on**

**"Pyrocumulonimbus Events over British Columbia in 2017: The Long-term Transport and Radiative Impacts of Smoke Aerosols in the Stratosphere" by Das et al.**

We thank the referees for their thoughtful reviews of our manuscript. Your insights have strengthened this work and we greatly appreciate your time and effort in doing so. We have made most of the recommended changes as described in details below, and the changes are reflected in the revised manuscript (in red ink) that is being submitted along with.

Please note that the original comments of the reviewers (RC) are in black and normal font while responses of the authors (AR) are in red and italicized. The line numbers in authors' responses refer to the line numbers of the revised manuscript unless specified otherwise. Also, typos and minor errors have been corrected based on the reviewer suggestions. Below are the detailed comments that needed addressing:

**Anonymous Referee #1**

**RC:** line 344 to the end of the page does not seem correct/logical at this point. You want to justify why OMPS and GEOS see enhanced aerosol extinction values at those 'high' altitudes, which is not so much related to the BDC. This (or similar) is what you could write: During the Asian summer monsoon tropospheric trace gases and aerosols are convectively lifted into the UTLS, where they remain largely confined within the transport barriers of the Asian monsoon anticyclone during the monsoon season (June-September) (references). During August 2017 the center of the ASMA was between 15-45N and 40-110E as defined in Kloss et al., 2019. This explains...

For the references here: I would not use Randel et al., 2010: They quickly mention the tracer enhancement within the ASMA, but the focus of this paper is about what happens after the break down of the ASMA (September/October), i.e., where do those 'isolated' air masses go. And even this mechanism has been extensively discussed and revised since then. Only a very small fraction will end up in the ascending branch of the BDC, which would not show in your data. I would use Park et al., 2008 (as an early paper) https://acp.copernicus.org/articles/8/757/2008/ or/and Santee et al., 2016 (a comprehensive more recent study) https://agupubs.onlinelibrary.wiley.com/doi/full/10.1002/2016JD026408 Both those studies focus on the trace gas enhancement within the ASMA.

Rather than citing Vernier 2015, I would take the original Vernier 2011 paper (where the ATAL: aerosolenhancementwithintheASMAwasfirstdiscovered):https://agupubs.onlinelibrary.wiley.com/doi/full/10.1029/2010GL046614

AND for your specific case in 2017 have a look at Kloss et al., 2019. They show the ATAL signal in 2017 in mid-August up to 18 km altitude with SAGE III data.

**AR:** Thank you for the clarification and excellent suggestions. We have revised this section 3.3 accordingly.**

**RC: Sentence starting in line 354:** No, this is not the ascending branch of the BDC, but rather a result of the circulation around/above the ASMA. Please have a look and refer to Wu et al., 2010: https://acp.copernicus.org/articles/17/13439/2017/

And Tissier and Legras, 2016: https://acp.copernicus.org/articles/16/3383/2016/

This feature is also seen in Vernier et al., 2011 (have a look at their Figure 3 e.g.), and again have a look at and refer to Kloss et al., 2019. There you even see for your specific example that the fire plume is around 2-3 km higher than the ATAL (their Figure 2). Also, mean SAGE III profiles at roughly the same time/region seem 1-2 km lower in altitude than what you observe with OMPS. The SAGE III profile matches the model results better. Maybe this might be worth mentioning?

**AR:** The section 3.3 is completely revised in light of your present and previous remarks. Regarding better matching of model plume top heights with SAGE-III profiles over ASMA region, we have added a sentence stating this in lines 384-387.

**RC: Sentence starting at 373:** This is a feature often seen after volcanic eruptions as well. You could cite Haywood et al. 2010 (https://agupubs.onlinelibrary.wiley.com/doi/full/10.1029/2010JD014447) for OSIRIS-HadGEM2 comparison after the Sarychev eruption (their Fig. 5) and Kloss et al., 2021 (https://acp.copernicus.org/articles/21/535/2021/) showing the same feature with OMPS data (compared to WACCM) for the Raikoke eruption (2019).

*AR:* We agree and thanks for pointing this out. We have revised the text accordingly in lines 400-403. See our response to Reviewer#2 about the same.

**RC:** Section 3.6: Please compare your radiative forcing values with the ones from Kloss et al., 2019 (UV Spec simulation based on SAGE III data in the ASMA region), they seem to match well. Furthermore (see also line 480), instead of or additionally to the Pinatubo comparison it might be worth bringing the radiative forcing estimations of your study from the BC fires in context to the most recent (much stronger) Australian fires. Please have a look at Khaykin et al., 2020 (https://www.nature.com/articles/s43247-020-00022-5). If

you'd like to compare your radiative forcing results to all stratospheric aerosol events of the past few years, also have a look at Kloss et al., 2021 (Figure 8 and 9 for radiative forcing estimations of the Raikoke plume).

**AR:** Based on both your and Reviewer#2 suggestions, we have now included a new Table 1 comparing our mean (clear-sky) radiative forcing estimates with those from Australian fires (Khaykin et al., 2020), as well as with the same BrCo pyroCb smoke aerosol impact over the ASMA region, as in Kloss et al. (2019). This is Table 1 of revised version. The accompanying discussion is added in section 3.6, lines 443-469.

**RC: Section 3.7 and general**: The model from Yu et al., 2019 was not nudged after 12 August 2017 (see their supplements), which limits the trustworthiness of the early distribution. Might this be worth considering when comparing your models and to explain some differences?

**AR:** Thanks for the suggestion, and we do include a sentence emphasizing that nudging (or replay in the context of GEOS) was crucial to our solution in section 4 and lines 518-520. However, it is hard to comment on the accuracy of the models in simulating the observed extinctions in the early phase because both observations (SAGE-III and OMPS-LP) have issues either in coverage or accurately measuring the aerosol extinctions during this time. Also, the two model results are probably closer to each other than the respective observations during this period. In terms of the horizontal distribution, comparing Figure 1C of Yu et al. (2019) to our Figure 8b, we agree that their aerosol appears to move more rapidly towards the pole compare to SAGE-III observations, while our simulation has more of the opposite.

**RC: Figure 4:** Could you clarify if those lines are averages over the whole NH (as for the background aerosol extinction)? But if so: Even though you only consider data above the cloud top height for OMPS, would the result not be biased by different tropopause altitudes (tropics vs. higher latitudes)? How can you distinguish between 'plume clouds' and other clouds for the OMPS observations?

*AR:* Thanks again for a great observation. We would like to clarify two things here: (1) Figure 4 lines are averages over 30-90°N instead of the complete NH. We clarify this now in the revised Figure caption as well as in the text. So, the tropopause altitude is mostly below 12 km over most parts of 30-90°N and the fact that we define the plume top as the first level from top of the atmosphere where we detect elevated aerosol layer will almost ensure it is always above the tropopause, and hence not a cloud. Figure 4 is also consistent with Figure 6 (and Fig. S1a-c), which show the location of the individual plumes in the NH, well above the tropopause, effectively confirming that it is not cloud contaminated. (2) We also clarify here that the OMPS-LP data we reported is actually (cloud) unfiltered data that was produced to ensure that potential biomass and volcanic aerosol signals are maintained in the stratosphere. This aspect has also been clarified in lines 202-205 under section 2.2.

**RC:** Minor comments/suggestions:**

Line 1 Erasing the term 'Pyrocumulonimbus events' would make the title more accessible and the pyroCb event itself is also not really the topic of your manuscript. All together your paper is a modeling study with satellite comparisons, I would add the term 'model study' or 'GEOS' in the title. Just a suggestion..

**AR: The manuscript title is slightly changed now.**

Line 19-21: 'The model simulated... are in close agreement'?

**AR: We believe this statement is correct as discussed in the paper.**

Line 50: You could include and generally also have a look at Lestrelin et al., in ACPD: https://acp.copernicus.org/preprints/acp-2020-1201/

**AR: Included.**

Around line 51: You might want to add ground based observations as well, e.g. Ansmann et al., 2018: https://acp.copernicus.org/articles/18/11831/2018/ Baars et al., 2019, Khaykin et al., 2018..

**AR: Done.**

A lot of the literature within the paper is not found in the reference list: e.g. Peterson et al., 2016; De Laat et al., 2012; Chen et al., 2016; Chen et al. 2020 ...

**AR: Done.**

Line 91/93-100: Don't you think the model description is a bit too detailed for the introduction? You might consider shifting it to the methods section.

*AR:* We agree that introduction might appear a little too detailed, but we would like to keep it the way it is to emphasize the motivation and uniqueness of our study upfront.

Line 161: The 'baseline experiment includes the injection'....?

**AR: Replaced 'baseline' with 'main'.**

Section 2.2: The newest OMPS version is 2.0, giving aerosol extinction observations on multiple wavelengths. I am wondering, why you didn't use the latest version.

*AR:* Most of the work for this study was completed before the processing of OMPS version 2.0 data was completed. Also, the version 2.0 validation paper was not yet published when we submitted our original manuscript.

Section 2.3: Why did you choose to bring SAGE III and CALIOP in one methods section?

**AR:** The simpler reasoning is that apart from OMPS-LP extinctions, which we majorly use for our model calibration and evaluation, we simply wanted to club together the other satellite observations that we briefly use under a single section.

Section 2.3, SAGE III: For OMPS you explain how you treat clouds, how about SAGE III? I understand that you only show one profile and the AE shows very obviously that there were no clouds at that point, but you also write that this is only an example.

**AR:** As defined in section 3.1, Angstrom Exponent (AE) relates inversely to the average size of the particles. For a cloud or very large particles, the AE is nearly zero and the extinction coefficient does not change with wavelength. We exclude SAGE III measurements where AE is zero. We have added this explanation in lines 241-242 in section 3.1.

Throughout the manuscript you change between 'SAGE III', 'SAGE-III' and 'SAGE-III/ISS'. I think the last version is the best..

**AR:** We keep SAGE-III nomenclature throughout when referring to the instrument, but add the ISS part when talking about the satellite overpass. Other discrepancies regarding this have been removed.

Line 239: 42°N seems too north to be called 'sub-tropical'

**AR: Corrected.**

Line 245: Maybe you could add a '(not shown here)'

**AR: Added.**

Line 323-328: Nice comparison. Maybe you could state the motivation of showing this Figure. In my eyes, this paragraph seems a bit lost at the beginning of 3.3.

AR: Thanks, we have now revised the entire section to hopefully make more sense.

Line 338: nor -> neither?

**AR: Corrected.**

Line 391: 'It is clear from the comparison,..., (that?) the magnitudes...'

**AR: Corrected.**

Line 397: You mean 'down to' I assume ..

**AR: Corrected.**

Line 412: compared

**AR: Corrected.**

Line 442: I thought you wrote earlier (section 3.3 / Figure 4) that the lofting around the ASMA is not well represented in the model.

AR: This part has been revised.

Line 466: August 2017 over..

AR: Corrected.

Line 477: that-> than

AR: Corrected.

In Data Availability: You might include the data version for SAGE III and CALIOP

AR: Added.

Figure 1: Could you replace the titles by more meaningful ones?

*AR*: It was not fully clear what is being meant by figure titles, but we did change the caption and the caption title to make it more meaningful.

Figure 6a: km-1

AR: Done.

**Anonymous Referee #2**

**RC:** My major concern is that the authors take no account for the underlying clouds that could significantly enhance the heating of stratospheric smoke (e.g. Boers et al., 2010). This issue is not even mentioned in the paper, which is surprising given the effort the authors invest to accurately constrain the smoke optics. This shortcoming should lead to underestimation of the diabatic ascent rate, which actually seems to be the case judging from Fig. 4, where the observed plume top appears 2 km higher than the simulation after T+20 days. Obviously, a global-scale simulation including the clouds would substantially complicate the modeling experiment. However, it would be of great value for this study if the authors estimate how much a convective cloud extending up to the tropopause layer (which were widely encountered in the ASMA region during that time) would affect the heating of the stratospheric smoke plume.

**AR:** We think this is a misunderstanding, possibly stemming from our lack of description about model clouds in section 2.1. We do have clouds in the simulation, and the aerosols are radiatively interactive with the clouds in the model. We clarify this in the revised version in lines 155-161 under section 2.1. We have also included an additional plot/panel in Figure 12b showing the difference between (originally reported) all-sky and clear-sky values of SW heating rates. The positive values in Fig. 12b are consistent with the understanding that dark aerosols over bright clouds lead to enhanced aerosol heating in the atmosphere, and hence verify the impacts of radiative interactions between stratospheric aerosol plumes and the underlying cloud for our case.

**Specific remarks**

Figure 2. It would be interesting to see the simulated and observed profiles in the same plot

AR: We have now modified Figure 2 as suggested.

Figure 3a. Would it be possible to display the peak values from Leipzig observation?

*AR:* As suggested, we now display the peak extinction values from Leipzig lidar observations in Figure 3a. We also include Figure 3b to explain the differences between the model simulated and observed extinctions, and discuss this in lines 271-288.

**RC: Figure 6.** The amount of information conveyed by this figure does not justify 40 panels. I believe most of them could be moved to supplementary material. I also have a concern on how the OMPS-LP data are presented in this figure: there appears to be a strong boxcar smoothing in the zonal direction, which was applied to mask the gaps between the orbits. It might be a better solution to aggregate OMPS-LP data over 3 days to avoid smoothing across these gaps.

**AR:** There is no smoothing applied to the OMPS LP data. The OMPS LP three slits were aggregated into daily file using  $1.5^{\circ}$  latitude x  $20^{\circ}$  longitude grids, effectively filling the whole longitude range. The advantage of one day vs 3 days gridding is that it provides more accurate tracking of the plume ascent into higher altitudes and better evaluation of the model's simulations. However, we do agree with the suggestion to move some of the sub-plots to supplementary material. So, we now keep only original Figure 6a and 6d in the main text and modified the discussions accordingly in section 3.3.

RC: Figure 7. The enhanced SAOD in the Arctic after day 150 is obviously due to PSCs. This should be mentioned in the text and/or figure caption. Alternatively, the PSCs could be removed using the same approach as in OMPS-LP V2.0 retrieval, which would enable a better comparison with the simulation.

**AR:** We agree that this enhancement is due to PSCs, and thanks for pointing this out. We have now acknowledged it in the caption of Figure 7.**

**RC:** p.10, l. 340. "... the model overestimates the background aerosol extinctions...". First, it is not totally obvious from Fig 6a that the model actually shows significantly higher extinctions. Secondly, the strongly enhanced extinction in the tropics is most certainly due to TTL cirrus and not due to aerosols.

**AR:** We agree that for OMPS, enhanced extinction in the tropics (at 16.5 km levels) might have contributions from TTL cirrus, but for the model we are plotting only the aerosol extinction fields. We have modified this part anyways based on previous suggestions, which should take care of this.

**RC:** p.10, l. 344 - 348. This sentence contains several statements that are either overly general or not entirely correct. The references provided are not quite relevant too. I would suggest to omit this sentence.

**AR:** These sentences have been revised with more appropriate references based on Reviewer#1 suggestions.

P.11, 1. 355 – 365. I disagree with the interpretation provided. The BC smoke plumes were mostly contained in 3 bubbles confined by smoke-charged vortices (SCV), of which only one (Vortex A in Lesterlin et al., 2020 https://acp.copernicus.org/preprints/acp-2020-1201/acp-2020-1201.pdf) ascended to 23-24 km, whereas the two others (B1 and B2) ones did not rise as high. The vortex A was already at 19 km while overpassing Europe (Khaykin et al., GRL, 2018) and by the time it arrived to the Asian region, it was already at 21 km, i.e., well above the Asian anticyclone (Lesterlin et al., 2020; Bourassa et al., JRG, 2019). Hence, the AMA could not have played much role in terms of the upward transport. Likewise, the cloud scavenging is of no relevance here as the A bubble is well above the TTL clouds. What could be the actual reason why the model falls short reproducing the diabatic rise is the absence of clouds in the simulation.

*AR: We have revised this section 3.3. to clarify the confusion related to ASMA based on both your and Reviewer#1 suggestions. Also, the cloud scavenging part is omitted.*

**RC:** p.11, l. 375 - 377. Given the large extent of the initial cloud, it is unlikely that it was located in between the 3-slit swaths. I believe, a more important reason why OMPS-LP doesn't get the early cloud is the saturation at extreme extinctions.

**AR:** We don't believe that the initial aerosol extinction levels were large enough to saturate the instrument. For example, the OMPS-LP algorithm had no problem retrieving the aerosol extinction values of Calbuco or Raikoke eruptions, which were larger than this event. In fact, OMPS first observation of the plume was on August 13 at 12 km, albeit the extinction value wasn't too large. It's quite common for limb instruments to underestimate the initial plume magnitude, as noted by Reviewer #1, Haywood et al., (2010), and Kloss et al., (2021). This is mostly caused by a combination of its coverage, when the plume is not well mixed, and OMPS LP large sampling along the line-of-sight, which is 125 km along-track, and up to 200 km cross track. We have revised this part in lines 400-403.

**RC:** p.11, l. 378. Same issue here. The injected smoke cloud is already above the tropopause on 14 August (Fig, 5b) therefore since the cloud screening was applied for clouds below the tropopause (Sect. 2.2), it should not have discarded this plume.

**AR:** Firstly, we would like to clarify here that the OMPS-LP data we reported is actually (cloud) unfiltered data that was produced to ensure that potential biomass and volcanic aerosol signals are maintained in the stratosphere. This aspect has now been clarified in lines 202-205 under section 2.2. Also, we agree with you that this means our initial explanation for this part was flawed, which has been replaced using the above stated argument. Figure 7a clearly shows similar aerosol distribution in the early days of this event to the model (Figure7b), just not quite as high as the model.

**RC:** p. 11, l. 381 and p.12, l.392. Total AOD would imply the entire column. Do you mean stratospheric AOD here?

AR: Total AOD -> stratospheric AOD

p.12, 1.387. rather be said "...there were no strong volcanic eruptions or PyroCb events..."

AR: This has now been corrected in lines 408-10.

p.12, 1.404. The sentence needs revision, cf. previous remarks.

AR: This has been revised based on the previous explanations as well.

**RC:** p.13, 1.423. It would be very useful to provide the value of global-equivalent monthly-mean RF to be compared with that of Australian bushfires (Khaykin et al., 2020).

**AR:** Based on both your and Reviewer#1 suggestions, we have now included a new Table 1 comparing our mean (clear-sky) radiative forcing estimates with those from Australian fires (Khaykin et al., 2020), as well as with the same BrCo pyroCb smoke aerosol impact over the ASMA region, as in Kloss et al. (2019). This is Table 1 of revised version. The accompanying discussion is added in section 3.6, lines 443-469.

**RC:** p.13, 1.425 - 427. The gaseous composition of the stratospheric PyroCb plumes is vastly different from the background values (strongly enhanced H2O and CO, depleted O3), which may potentially have an important impact on the plume heating. This should be discussed in a more careful way, i.e. how the alteration of these gases could affect the plume rise.

**AR:** We apologize for the confusion (original) lines 425-427 might have caused and thanks for pointing this out. We want to clarify here that even though we do not explicitly demonstrate or quantify the impacts of enhanced water vapor and depleted ozone on plume heating, these impacts should have been nonetheless accounted for within the model since we do have the StratChem chemistry module (that includes stratospheric ozone chemistry) coupled to the radiative transfer scheme within the global model. Additional CO emissions due to pyroCb were not included, but this should not impact plume heating either since it is not a radiatively active gas in the model. As we mentioned in Section 2.1 of model description, we performed several simulations with varying model set-ups before arriving at the "best-estimate" simulation, and in our test simulations, we found that the resultant impact of both, enhanced water vapor and depleted ozone in the stratosphere, led to net radiative cooling of the atmosphere and thus provided a negative feedback on the plume rise (also shown in Oman et al. (2018), presented at AGU Fall Meeting, December 2018, https://agu.confex.com/agu/fm18/meetingapp.cgi/Paper/409405). We would also like to note that, both enhanced water vapor and depleted ozone within the pyroCb plumes, as observed by satellite instrument (e.g., MLS), occurred majorly due to the dynamical perturbations from the rapid diabatic rise of the heated plume from the tropopause through the lower stratosphere. Since we allowed for temperature and water vapor (Ov) blending below the tropopause levels in our final model set-up (see Section 2.1 for details), the model was able to largely reproduce these observed changes as well. We do not explicitly demonstrate the results here, but we acknowledge that these could be interesting findings to report in a separate study.

Based on this suggestion, we have modified and included additional discussion in lines 470-78 under section 3.6 in the revised manuscript.

**Technical remarks**

RC: p.4, l. 114, In August *AR: Corrected.*RC: p.6, l.196. The reference to Chen et al. is missing in the bibliography. *AR: Included now.*

**Anonymous Referee #3**

**Major comment:**

**RC:** The authors have optimized their model for aerosol size by using the Angstrom Exponent (AE) derived from SAGE III retrievals, but I didn't find any explicit discussion of the possible impact of morphology or shape of the aerosol particles. In particular, one of the most intriguing observations from the Canadian pyroCb events in August 2017 as well as from the Australian events of January 2020 is the high depolarization ratio in the plumes implying significantly non-spherical particles. As shown by Christian et al. (2020), the depolarization ratio for the Canadian pyroCb kept increasing for several weeks after injection. Several explanations have been put forward in the recent papers but the issue is not completely settled, in my opinion. In any case, I believe realistic modelling efforts should address this aspect and its impact on the evolution of the plume. Yu et al. (2019) had specifically addressed this by using fractal aggregates of black carbon. They found that these fractal aggregates produce higher absorption in the mid-visible. I am wondering if inclusion of non-spherical particles will help solve the lower amount of lifting produced by the model in the current study as compared to the observations. I believe the paper would improve significantly by addressing this issue.

**AR:** We agree that there is still a large uncertainty associated with the assumptions of both morphology and optical properties of the smoke particles. Since GOCART assumes both BC and BrC components to be spherical particles and therefore non-depolarizing, we are fundamentally limited by our assumptions of particle shape in this regard. Definitely more sophisticated aerosol microphysics modules that allow aerosol absorption and size distribution to evolve with age, as well as more realistic morphology might help close the gap between model and observations in future studies. However, as you pointed out that while Yu et al. (2019) modeled their BC as fractal aggregates coated with non-spherical organics to incorporate the irregularity in shapes of the smoke particles, but still could not match the high values of depolarization ratios observed by the lidar instruments.

Regarding SSA, our study also tried to calibrate these properties to the best of our ability using groundbased lidars and SAGE-III observations (in section 3.1) despite the limitations of our relatively simpler aerosol microphysics. Of course, more constraining is required.

Further regarding SSA and associated lofting, we agree that our model is probably not able to loft the aerosols to the highest levels of smoke observed by OMPS, i.e., ~ 22 km. However firstly, lower SSA=0.85 for the (fractal) BC coated with OC reported in Yu et al. is the intensive property of the 2% of the smoke mixture, not for the entire mixture of 98% OC + 2% BC coated with OC. What we report as SSA=0.9 at 532 nm is for the stratospheric smoke mixture at ~15km over Leipzig. Secondly, if we look at Fig. 1A and

Fig. 3 of Yu et al., we don't see their aerosol plumes rising higher than  $\sim 20$  km either. So, it is difficult to implicate that the model falling short by 1-2 km during its final ascent is solely due to absorption assumption. We think that having higher model vertical resolution could also play a role, since the plume would be more confined and can lead to higher localized aerosol heating rates.

**Minor comments:**

**RC:** Lines 212-213: CALIOP measures both the parallel and perpendicular component of the backscattering signal at 532 nm but only the total backscatter at 1064 nm.

*AR: Thanks for pointing this out, we have corrected the related text accordingly.*

**RC:** Please use the same scale on the x axis for Figures 2a and 2b. In this figure the comparison between model and SAGE III extinction profiles shows a peak at  $\sim$ 14 km on September 3. Is this from another episode of pyroCb, since going by Figure 4, the plumes should be near 22 km by September 3?

*AR:* Based on both your and Reviewer#2 suggestion, we have modified Figure 2 now. The main plume broke into 3 major vortices (Listerlin et al., 2020) after the initial period up to about August 25-27. Notice that Figure 4 is tracking only the highest plume, not all of them. So, the example case in Figure 2 is most likely a part of one of the other vortices that did not rise as high. Also, see Fig. 6b, highest plume on this day is around  $45^{\circ}N$ ,  $90^{\circ}E$  and  $\sim 22$  km.

**RC:** It is not very clear to me if the simulated AE shown here was actually used or is it just to show the methodology used as an example, as the authors state, page 7, line 236? Would it not be better to present this comparison for some cases in August?

**AR:** SAGE III/ISS is an occultation instrument that is limited to 30 measurements a day. September is the earliest when SAGE III had a good view of this event. This SAGE III profile is just an example of a typical aerosol profile and the observed AE. The observed plume altitude and the observed AE varied depending on the location of the measurements, however, the observed AE mean was close to the shown value of 1.6 based on SAGE-III. For the model, we actually adjusted the BrC optics (mainly the modal radius and

particle size distribution) such that the resulting model AE were closer to SAGE-III. One example comparison for September 3 after the model adjustment/calibration is depicted in Figure 2.

RC: Why not show only the SAGE III extinction profile for the wavelength being compared at?

AR: This has been taken care of in revised Figure 2.

RC: Since the simulations run for several months for the aging smoke, I wonder if a time dependent AE and SSA were considered for these simulations.

**AR:** No, GOCART (aerosol module) does not have the more sophisticated aerosol microphysics to track the aerosol size distribution or SSA with plume evolution or age. The variation in SSA and AE in the model is only due to change in composition (i.e., due to different aerosol source and sink processes affecting a given aerosol volume) and due to the conversion between hydrophobic to hydrophilic fractions of BC and BrC and subsequent response to changes in relative humidity (less significant in the stratosphere).

4. The authors present a very brief comparison of the CALIPSO data and simulation in Figure 5. Firstly, why is the color bar for the CALIPSO backscatter placed upside down? It can be a little confusing. Also, why is there an apparent discontinuity near 8 km in both the panels? The scene shown in Figure 5a was also shown in Torres et al. (2020) but using the directly available CALIPSO browse image and this discontinuity was not seen in that figure. I suspect it is a plotting artifact and related to the vertical resolution of the downlinked data changing around this altitude. This should be clarified in the text for the benefit of readers. This plume observed by CALIPSO actually has interesting constraints for modelling. As mentioned in Torres et al. (2020), this plume (Figure 5a) is a mixture of ice and smoke, with rather high depolarization ratio of 0.2-0.5. While ice is not likely to survive in the stratosphere for too long, this is an ice-smoke mixture and it will be interesting to study if this mixture impacts the evolution of the plume in the first few days when the plume is rising steeply. In fact for the Australian pyroCb events of January 2020, Khaykin et al. (2020) have presented evidence of ice up to 22 km from MLS ice-water content retrievals. Also, I notice that the model missed out those parts of the plume which are more aerosol rich, i.e. the extended plume between 6-10 km between 60N-65N. This would have been clear if the authors had presented the attenuated color ratio image from CALIPSO.

**AR:** All of these are good points, and especially the part about the impacts of ice-smoke mixture on plume evolution is worthy of further exploration, but beyond the scope of this study. For this study, we focus more on the long-term transport and radiative impacts and primarily use OMPS-LP observations for model calibration. Regarding the extended plume part between 6-10 km, we acknowledge that since we don't fully resolve convective activity at the scale of fire events and instead rely on direct injections of the stratospheric

smoke component (in the mode), we miss some of the vertically lifted smoke that remains in the troposphere. However, this would not change the conclusion of our paper.

Lastly, we do clarify the discontinuity in CALIOP figures around 8 km (in the caption of Figure 5), which is indeed related to the change in vertical resolution of the downlinked data around this altitude. Thanks for the suggestion.

RC: I wonder why the authors did not use the latest version 2.0 OMPS data. Perhaps some of the differences between the model and OMPS data noted by the authors could be alleviated by the new data.

*AR:* Possibly so, but we would like to clarify that most of the work for this study was completed before the processing of OMPS version 2.0 data was completed. Also, the version 2.0 validation paper was not yet published when we submitted our original manuscript.

RC: Section 3.3 on the link to the ASMA is not particularly convincing. For instance, referring to Figure 6a, I am not sure if I understand the statement on line 347-348. The enhanced values of extinction at 16 km and even at 18 km over south Asia seem to be simply the ATAL feature, which the authors don't seem to mention. Also in addition to these lat-lon plots, a height latitude plot of the extinction might be useful. The legend for Figure 6a may need re-wording, since no smoke is seen in either model or OMPS data in this figure. Similarly the statements in lines 355-360 are not clear or convincing.

*AR:* Based on your and previous remarks of Reviewer#1 and 2, we have revised this entire section to reflect most of what you are suggesting.

RC: Figure 9—I am curious as to why there is a bump in AOD around day 180 after the pyroCb injection with the model showing a significant underestimate particularly at 16 km.

*AR:* The sharp increase is mostly due to the presence of polar stratosphere clouds (PSCs) in the OMPS-LP data, which we now clarified in the caption of Fig. 7.

RC: In Figure 11a, why is there a sharp drop from the peak value to about day 10 and then a slow rise to about day 30 before the steady decline?

*AR:* This has again to do with the breaking up of the initial large plume due to vertical shear (Listrelin et al., 2020). Based on Listrelin et al. and also in our model, a part of the plume separated from the main

plume (about Aug 16-19), which remained closer to the tropopause and gradually disappeared, possibly returning back to the troposphere as the plume moved southward to higher tropopause heights.

RC: Lines 454-455—since the 2 estimates differ by such a lot (factor of 20-25), is there any way to evaluate this?

AR: As we explain in the text (lines 502-10), this is most likely due to the differences in our injected BC mass. Regarding evaluation, maybe it will help now that we compared our TOA and surface forcing estimates to other studies on BrCo pyroCb (e.g., Kloss et al. 2019) in Table 1, and our results show comparable magnitudes to the previous study. Therefore, providing more confidence in our reported SW heating rates that are bounded by the TOA and surface forcing.

---

## Author Comment (AC2) · 27 Apr 2021

**Response to Short Comment on "Pyrocumulonimbus Events over British Columbia in 2017: The Long-term Transport and Radiative Impacts of Smoke Aerosols in the Stratosphere" by Das et al.**

Dear Dr. Albert Ansmann,

Thank you very much for posting your insightful and important comments on the discussion forum. Please see our detailed response below and the related changes are reflected in the revised manuscript (in red ink) that is being submitted along with.

[**SC**]: Short comment
[***AR***]: *Author Response in Italic*

[**SC:**] This paper will become an important contribution to the stratospheric smoke literature! That motivated me to write this comment. Baars et al. (ACP, 2019) presented a dense set of lidar network information on geometrical, optical and microphysical properties of the stratospheric smoke over Europe after the strong pyro-CB-related smoke event of August 2017. You mention the paper briefly in your article. The paper covers six months of smoke observations!
The Baars et al paper should be mentioned already in the introduction as it is an important observational contribution to the research and documentation of the recordbreaking smoke event, that you are modelling. Furthermore, the European lidar network results should then be compared with your model findings (for Europe).
I am curious to see how your model results agree with this height-resolved smoke lidar data set!
How well do the model results agree with the lidar data in terms of optical depth or even layer-mean extinction coefficient?
Does the model resolve properly the height range of smoke observed over Europe, from Northern Norway to southern Portugal and Spain (western Mediterranean) and Cyprus and Israel, in the Eastern Mediterranean.
To be more precise:
Figure 2: Why did you not use the Baars-et-al.-2019 data (although knowing this paper
and the results) in the comparisons shown here?
Figure 3: Here, you use lidar data from Europe (even from Leipzig!) Very good, thank you!
Figure 4: Here, it would make really sense to take the European lidar data (on smoke
layer top heights) to check the quality of the model results.
Figure 6c, 6d, 6e: another excellent opportunity for lidar (Baars et al.) vs model comparisons,...with Europe in the center of all your plots.
Finally, Figure 7 and Figure 8 results should or could be compared with the extinction coefficients presented by Baars et al. (2019).

[*AR*]: *We acknowledge the important contribution that Baars et al. (2019) has made in providing the detailed lidar observations for the smoke event of interest. We have now emphasized this further in our revised manuscript in the introduction section (section1, lines 50-52) as well.*

*We have also included additional details in section 3.1 (revised Fig. 3a, new Figures 3b, 3d) following some of your suggestions above. Most importantly, we attempted to show the mean and maximum of stratospheric AOT over Europe from our model (Fig. 3d) that would be comparable to Fig. 4b of Baars et al. (2019). The associated discussion is added in lines 297-304 under the same section.*

*Regarding your remaining comments, we agree they are excellent suggestions and worthy of further analysis, but is beyond the scope of this paper, which studies the long term and global impact of the BrCo pyroCb event. Your suggestions are certainly worth pursuing in a separate paper that specifically compare the model results to the lidar network measurements over Europe.*

---

## Author Response (AR2)

"The Long-term Transport and Radiative Impacts of the 2017 British Columbia Pyrocumulonimbus Smoke Aerosols in the Stratosphere" by Das et al.

Dear Editor,

We thank you for your careful reading of our revised manuscript and nice suggestions. Both, your and reviewer comments, have strengthened this work and we greatly appreciate your time and effort in doing so. We have made the recommended changes as described below:

*Editor comment: The authors state in abstract "The plumes resided in the lower stratosphere for about 8-10 months following the fire injections." Based on Figure 11 and discussion in text - I suggest modifying this to "Plumes were observed in the lower stratosphere for 8-10 months and simulated by model with a stratospheric e-folding time of about 5 months" to be accurate. Also, I would like a clear explanation of the structure Figure 11 - the initial rapid rise from direct (injection) followed by a steep fall (?) followed by a gradual rise and then a steady decay over 140 days. Also, it would be valuable to see data overlaid on Figure 11.*

Author response:

1. Abstract *lines 13-15* have been changed to include the e-folding time.
2. Fig. 11 has been appropriately modified to overlay the OMPS-LP data. The figure now shows model mass on primary y-axis (as before) and OMPS-LP retrieved AOD on secondary y-axis, along with their respective e-folding times.
3. The initial abrupt period in the model mass curve has been explained in *lines 495-510* of section 3.7, where the figure belonged.

Apart from the suggested revisions, we modified the Acknowledgement section to reflect our appreciation for the editor and the reviewers' contribution.

Thanks!

Sampa Das and co-authors